# Daxx inhibits hypoxia-induced lung cancer cell metastasis by suppressing the HIF-1α/HDAC1/Slug axis

Ching-Wen Lin[1], Lu-Kai Wang[2], Shu-Ping Wang[3], Yi-Liang Chang[4], Yi-Ying Wu[5], Hsuan-Yu Chen[6], Tzu-Hung Hsiao[7], Wei-Yun Lai[8], Hsuan-Hsuan Lu[9], Ya-Hsuan Chang[6], Shuenn-Chen Yang[1], Ming-Wei Lin[10], Chi-Yuan Chen[11], Tse-Ming Hong[5] & Pan-Chyr Yang[1,9]

Hypoxia is a major driving force of cancer invasion and metastasis. Here we show that death domain-associated protein (Daxx) acts to negatively regulate hypoxia-induced cell dissemination and invasion by inhibiting the HIF-1α/HDAC1/Slug pathway. Daxx directly binds to the DNA-binding domain of Slug, impeding histone deacetylase 1 (HDAC1) recruitment and antagonizing Slug E-box binding. This, in turn, stimulates E-cadherin and occludin expression and suppresses Slug-mediated epithelial–mesenchymal transition (EMT) and cell invasiveness. Under hypoxic conditions, stabilized hypoxia-inducible factor (HIF)-1α downregulates Daxx expression and promotes cancer invasion, whereas re-expression of Daxx represses hypoxia-induced cancer invasion. Daxx also suppresses Slug-mediated lung cancer metastasis in an orthotopic lung metastasis mouse model. Using clinical tumour samples, we confirmed that the HIF-1α/Daxx/Slug pathway is an outcome predictor. Our results support that Daxx can act as a repressor in controlling HIF-1α/HDAC1/Slug-mediated cancer cell invasion and is a potential therapeutic target for inhibition of cancer metastasis.

[1] Institute of Biomedical Sciences, Academia Sinica, Taipei 11529, Taiwan. [2] Radiation Biology Core Laboratory of Institute for Radiological Research, Chang Gung University/Chang Gung Memorial Hospital, Taoyuan 33305, Taiwan. [3] Laboratory of Biochemistry and Molecular Biology, The Rockefeller University, New York, New York 10065, USA. [4] Department of Pathology and Graduate Institute of Pathology, National Taiwan University College of Medicine, Taipei 10051, Taiwan. [5] Graduate Institute of Clinical Medicine, National Cheng Kung University, Tainan 70101, Taiwan. [6] Institute of Statistical Science, Academia Sinica, Taipei 11529, Taiwan. [7] Department of Medical Research, Taichung Veterans General Hospital, Taichung 40705, Taiwan. [8] Aptamer Core, Institute of Biomedical Sciences, Academia Sinica, Taipei 11529, Taiwan. [9] Department of Internal Medicine, National Taiwan University College of Medicine, Taipei 10051, Taiwan. [10] Program in Molecular Medicine, National Yang-Ming University and Academia Sinica, Taipei 11221, Taiwan. [11] Graduate Institute of Health Industry Technology and Research Center for Industry of Human Ecology, College of Human Ecology, Chang Gung University of Science and Technology, Taoyuan 33303, Taiwan. Correspondence and requests for materials should be addressed to T.-M.H. (email: tmhong@mail.ncku.edu.tw) or to P.-C.Y. (email: pcyang@ntu.edu.tw).

Emerging evidence has shown that the hypoxic nature of the tumour microenvironment is closely associated with late-stage cancer progression and metastasis[1,2]. Under hypoxic conditions, the hypoxia-inducible factors (HIFs), HIF-1α and HIF-2α, are stabilized, enabling them to coordinately regulate the expression of genes required for promoting disseminated, invasive and angiogenic properties, shifting cancer cells towards a metastatic phenotype[3,4]. Specifically, hypoxia-stabilized HIF-1α has been shown to upregulate epithelial–mesenchymal transition (EMT)-related transcription factors (EMT-TFs), including TWIST and Snail, indicating that HIF-1α plays a critical role in hypoxia-induced EMT[5,6]. In addition, inhibition of HIF signalling pathways improves clinical outcome in patients with renal cell carcinoma and oesophageal squamous cell carcinoma[7–9]. However, the molecular mechanism by which hypoxia impacts lung cancer metastasis is incompletely characterized.

Metastasis, a crucial determinant of cancer-related mortality, is initiated by a process in which tumour cells disseminate and gain invasive ability, a step referred to as EMT[10]. Downregulation of epithelial cadherin (E-cadherin) and tight junction molecules, such as occludins and the zona occludens proteins, ZO-1/2, during solid tumour dissemination is recognized as a pivotal phenomenon that is tightly linked to cancer aggressiveness and patients' outcomes[11–13]. Slug, an EMT-TF, has been shown to transcriptionally suppress the expression of E-cadherin and occludin, and promote cancer invasion and metastasis in various types of cancers[14–17]. Previously, we showed that the Slug-E-cadherin axis is associated with cancer metastasis and clinical outcome in non-small-cell lung cancers (NSCLCs)[18,19], suggesting that Slug is critically involved in lung cancer progression. Thus, identifying factors that regulate the metastasis-promoting actions of the Slug-E-cadherin/occludin axis would be important for the development of therapeutic strategies to target cancer metastasis.

Daxx (death domain-associated protein) has been shown to directly interact with and suppress multiple transcription factors, including Etwenty six 1, glucocorticoid receptor, androgen receptor (AR), nuclear factor-κB, p53, E2F1 and Pax gene family members, and it is involved in multiple biological functions[20–25]. In addition, through interactions with chromatin-remodeling proteins, Daxx has also been found to associate with histones to alter gene transcription[26–29]. The dynamic interaction between Daxx and its associated proteins is tightly controlled and required for tissue and embryo development[30–32]. Hence, dysregulation of Daxx and its associated proteins can affect tissue development, as well as cancer progression[32–36].

In this study, we show that Daxx is a negative regulator of hypoxia-induced EMT and cancer metastasis that acts by inhibiting the HIF-1α/HDAC1/Slug pathway. By directly interacting with the Slug DNA-binding domain, Daxx antagonizes Slug E-box binding, thereby subsequently stimulating E-cadherin and occludin expression. This stabilization of E-cadherin and occludin expression by Daxx prevents cell dissemination, and thus suppresses cancer cell invasion and metastasis during hypoxia. Our results and clinical evidence indicate that Daxx is a potential therapeutic target in strategies designed to inhibit cancer metastasis.

## Results

**Daxx function as an invasion and migration suppressor.** Daxx has multiple roles in various biological processes and human diseases, including cancer[37,38]. To study the potential roles of Daxx in lung cancer invasion and/or metastasis, we first investigated endogenous Daxx expression in various lung cancer cell lines. Interestingly, we found that Daxx expression generally correlated with expressions of E-cadherin and occludin, and inversely correlated with cell invasiveness (Supplementary Fig. 1a,b), suggesting a potential role of Daxx in regulating cancer cell invasiveness.

To test whether Daxx is involved in regulating cell invasiveness, we knocked down endogenous Daxx using small interfering RNA (siRNA). Daxx knockdown significantly enhanced cell invasive and three-dimensional (3D) migratory abilities (Fig. 1a,b; Supplementary Fig. 2a), an effect that was associated with downregulation of E-cadherin and occludin, and upregulation of N-cadherin (Fig. 1c; Supplementary Fig. 2b). Conversely, ectopic expression of Daxx in lung cancer cells with high invasive ability (CL1–5 and CL–141 cells) significantly decreased cell invasive ability and 3D migratory ability (Fig. 1d,e). In addition, the epithelial markers E-cadherin and occludin were upregulated in Daxx-overexpressing cells (Fig. 1f). These results suggest that Daxx downregulates cell invasion through modulation of epithelial markers.

Previous studies have shown that Slug transcriptionally represses E-cadherin and occludin, thereby promoting EMT and cell invasiveness[18]. In lung cancer cells, the expression of E-cadherin and occludin was upregulated in Slug-knockdown cells, a relationship opposite that observed under Daxx-knockdown conditions, implying that Daxx effects Slug-mediated cell invasion (Supplementary Fig. 2c). However, modulation of Daxx expression level did not affect Slug expression (Fig. 1c,f). Instead, we found that E-cadherin and occludin messenger RNA (mRNA) levels were downregulated in the absence of Daxx (Fig. 1g), inspiring us to explore whether Daxx regulates their expression by modulating the transcription-suppressing activity of Slug.

**Daxx suppresses Slug-mediated transcriptional repression.** We first determined the subcellular localization of Daxx and Slug, demonstrating that both were mainly co-localized in the nucleus (Fig. 2a; Supplementary Fig. 3a). Co-immunoprecipitation assays performed in CL1–5 cells using anti-Daxx and anti-Slug antibodies further demonstrated reciprocal interactions between endogenous Daxx and Slug (Fig. 2b). A direct interaction between Daxx and Slug was further confirmed by *in vitro* glutathione S transferase (GST) pull-down assays (Supplementary Fig. 3b).

To determine whether Daxx is involved in regulating Slug-mediated gene repression, we used a promoter–reporter construct containing Slug-binding site reporter. The luciferase activity of this Slug-responsive reporter construct was repressed in the presence of Slug; notably, increasing the level of Daxx expression resulted in a dose-dependent attenuation of Slug-mediated repression of luciferase activity (Fig. 2c). To further investigate whether Daxx modulates Slug-mediated E-cadherin repression, we co-transfected cells with an E-cadherin promoter–luciferase reporter construct containing wild-type or mutant forms of the E-box sequence together with Slug and/or Daxx. Co-expression of Daxx significantly blunted Slug-mediated suppression of wild-type E-cadherin-reporter activity, whereas neither Slug nor Daxx alone affected the activity of the mutant E-cadherin-reporter (Fig. 2d).

To study the mechanism by which Daxx suppresses Slug-mediated gene repression, we performed *in vitro* electrophoretic mobility shift assays (EMSAs) using purified Daxx and Slug proteins (Supplementary Fig. 3c), and a wild-type E-box-containing probe. As shown in Fig. 2e, Slug formed a complex with the E-box probe and this DNA–protein complex was disrupted by the addition of Daxx protein, suggesting that the binding of Daxx to Slug interferes with the ability of Slug to bind DNA (Fig. 2e). Interestingly, Daxx alone didn't form complex

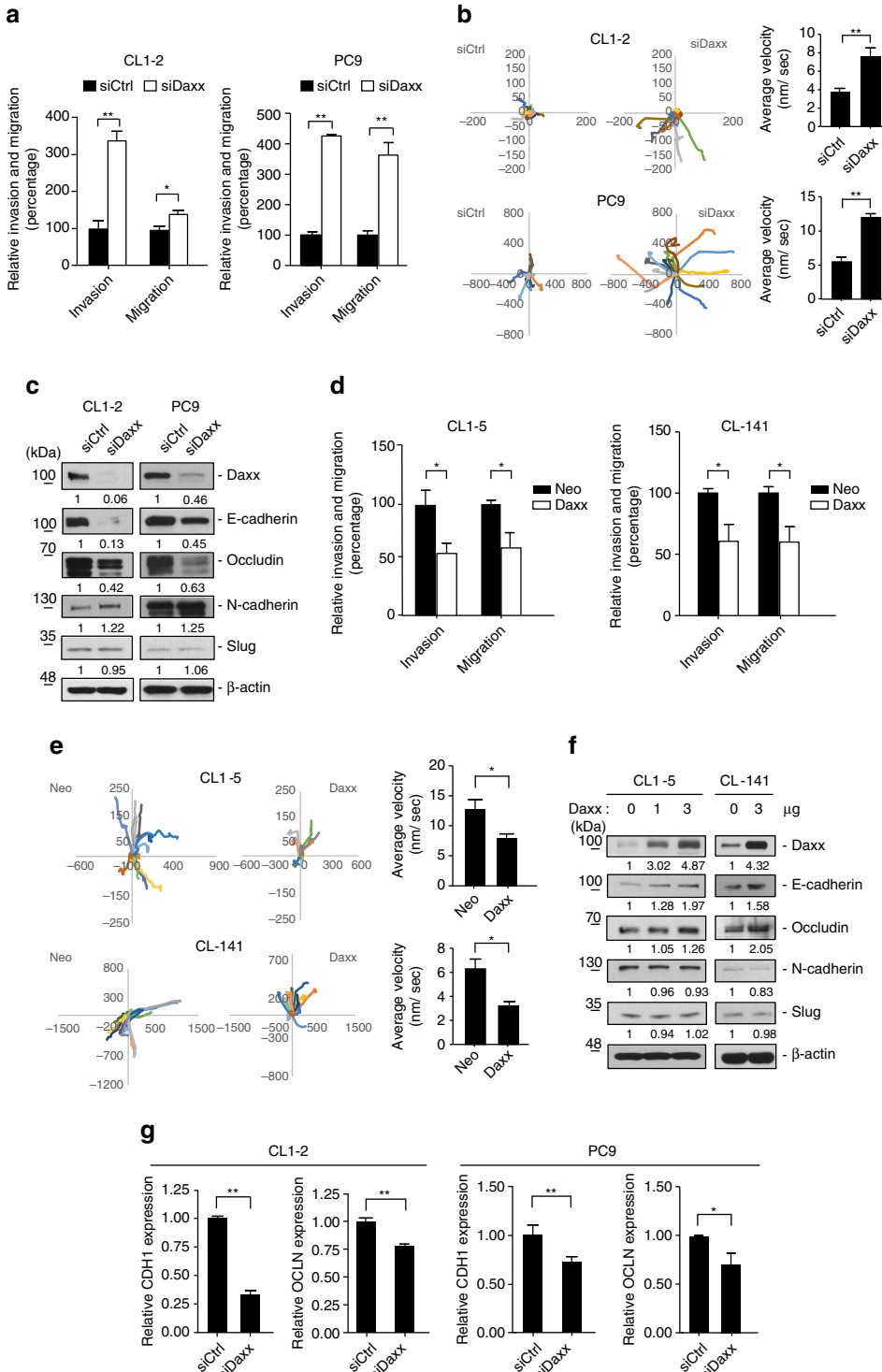

**Figure 1 | Daxx functions as an EMT and invasion suppressor.** (**a**,**b**) CL1–2 and PC9 cells were transfected with control siRNA (siCtrl) or Daxx siRNA (siDaxx). After 48 h, cells were used in invasion and migration assays. (**a**) Relative invasiveness are presented as percentage ± s.d. (n = 3). (**b**) 3D invasion assay using a single-cell tracking, time-lapse video microscopy system. Left panels: Representative track plots of individual control (siCtrl) and Daxx-knockdown (siDaxx) cells followed for 6 h. Right panel: quantification of average velocity of track plots (mean ± s.e.m. of 15 cells analysed in three independent experiments). (**c**) CL1–2 and PC9 cells were transfected with siCtrl or siDaxx. The indicated EMT markers were detected in whole-cell lysates by immunoblotting. (**d**) Invasion and migration assays were performed in CL1-5 and CL-141 cells transfected with control vector (Neo) or Daxx expression plasmid (Daxx). (**e**) 3D invasion assay were performed in CL1-5 and CL-141 cells overexpressed Neo or Daxx. Left panel: track plots of individual control (Neo) and Daxx overexpression (Daxx) cells followed for 6 h. Right panel: quantification of average velocity of track plots (mean ± s.e.m. of 15 cells analysed in three independent experiments). (**f**) CL1-5 and CL-141 cells were transfected with control vector or increasing doses of Daxx plasmids. Expression of the indicated proteins was assessed by immunoblotting. β-actin was used as an internal control. (**g**) CL1–2 and PC9 cells were transfected with siCtrl or siDaxx. Total RNA was isolated for quantification of E-cadherin (CDH1) and occludin (OCLN) mRNA levels by RT–qPCR. CDH1 and OCLN mRNA expression were normalized to that of GAPDH. Error bars represent s.d. *P < 0.05, **P < 0.01, paired two-way Student's t-test.

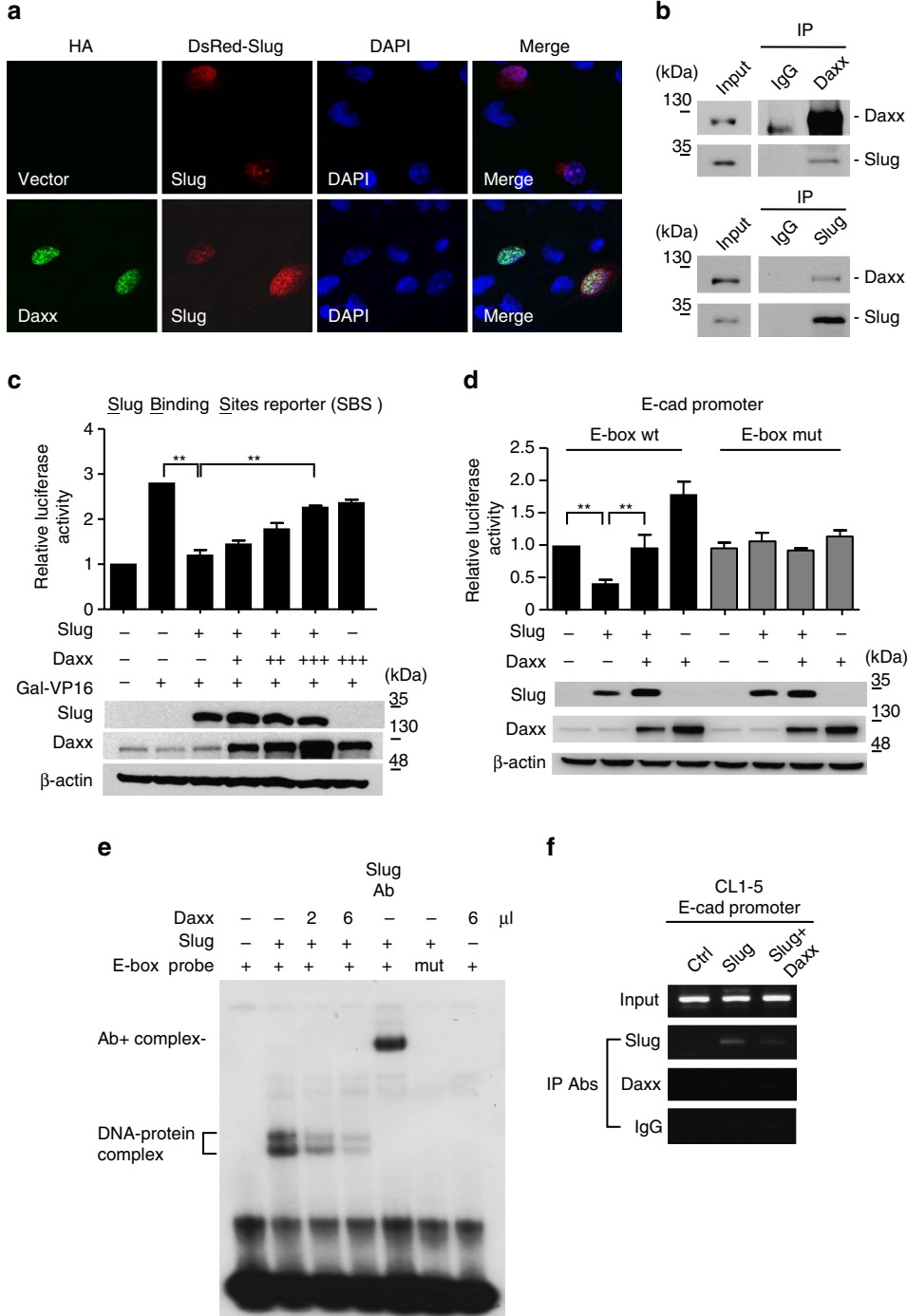

**Figure 2 | Daxx interacts with Slug and suppresses Slug-mediated transcriptional repression of E-box-containing genes. (a)** Localization of Daxx (green), Slug (red) and 4,6-diamidino-2-phenylindole (DAPI; blue) was analysed by immunofluorescence staining. H1299 cells were co-transfected with DsRed-Slug and HA vector or HA-Daxx. After 36 h, cells were fixed, immunostained with an anti-HA antibody and counterstained with the nuclear dye, DAPI. Fluorescence signals were obtained by confocal microscopy. **(b)** The interaction between Daxx and Slug was assessed by co-immunoprecipitation (co-IP) assay. CL1–5 cell lysates were immunoprecipitated with anti-Daxx, anti-rabbit IgG (upper panel), anti-Slug or anti-goat IgG (lower panel) antibodies, and immunoblotted with the indicated antibodies. **(c,d)** CL1-0 cells expressing the luciferase reporter plasmid 3 × SBS-Luc and the Gal4-VP16 activator **(c)** or wild-type (wt) or mutant (mut) E-cadherin promoter **(d)** were co-transfected with plasmids expressing Slug or Daxx. Luciferase activity was assayed 30 h later and normalized to that of Renilla (pRL-SV40), which serves as an internal control plasmid. Each data point represents the mean ± s.d. Experiments were performed twice in triplicate. **(e)** Effect of Daxx on the DNA-binding ability of Slug was assessed by EMSA. Wild-type or mutant E-boxC probes labelled with $^{32}$P were incubated with HA-Daxx or HA-Slug proteins, generated by *in vitro* translation, or with an anti-Slug antibody. **(f)** ChIP–PCR analyses of Daxx and Slug on the endogenous E-cadherin promoter in CL1–5 cells overexpressing Slug or both Slug and Daxx, performed using the indicated antibodies. **\*\*P<0.01**, paired two-way Student's *t*-test compared. This experiment used identical input and IgG control with Fig. 3c.

with E-box probe (Fig. 2e), suggesting that Daxx sequesters Slug from its target promoters. To test the possible role of Daxx in disrupting Slug DNA-binding ability, we performed chromatin immunoprecipitation (ChIP) assays. In CL1–5 cells, Slug protein was recruited to the E-cadherin promoter and the CAR promoter, another validated Slug target gene[39]; however, this recruitment was decreased by co-expression of Daxx (Fig. 2f; Supplementary Fig. 4). Collectively, these results indicate that Daxx suppresses Slug-mediated gene repression by impairing the DNA-binding activity of Slug.

**Daxx interacts with zinc-finger motifs of Slug.** To further examine whether Daxx suppresses Slug transcriptional activity through interactions with Slug, we generated a series of Slug deletion constructs. By co-expressing HA-tagged Daxx and various Flag-tagged Slug deletion constructs, we found that the C-terminus of Slug, which contains five repeated zinc-finger motifs, is required for Daxx interaction (Fig. 3a). Interestingly, the ability of Daxx and Slug to interact was steadily attenuated by decreasing the number of zinc-finger motif repeats in Slug (Supplementary Fig. 5a). To verify the Slug-interacting domain of Daxx, we also generated several Flag-tagged Daxx truncations, as illustrated in Fig. 3b. Both immunoprecipitation and GST pull-down assays showed that a minimum Daxx helical domain corresponding to amino acids 180–400 was required for the interaction with Slug (Fig. 3b; Supplementary Fig. 5b). These results suggest that Daxx antagonizes Slug DNA-binding ability by directly interacting with the Slug DNA-binding domain.

**Daxx interferes with Slug/HDAC1 to target sites.** Previous studies have shown that HDAC1 is required for Slug to execute its transcriptional repressor activity[18,40]. Using an anti-HDAC1 antibody in ChIP assays, we confirmed that HDAC1 could be recruited to E-cadherin and CAR promoter regions in the presence of Slug (Fig. 3c; Supplementary Fig. 4). Co-expression of wild-type Daxx eliminated the association between Slug and HDAC1 (Fig. 3d) and abolished HDAC1 recruitment (Fig. 3c; Supplementary Fig. 4). In contrast, a Slug-binding-defective Daxx mutant containing a truncated helical domain (designated DD1) was unable to interact with HDAC1 (Fig. 3e; Supplementary Fig. 6). These results suggest that Daxx inhibits the Slug/HDAC1 transcriptional repressor complex by functionally interfering with Slug–HDAC1 interactions.

**Daxx suppresses EMT and invasion through binding with Slug.** Because Slug is a key member of the family of EMT-inducing transcription factors, we next investigated whether Daxx affects epithelial markers and cell invasion by functionally repressing Slug. First, we determined whether endogenous Slug and Daxx affect the mRNA expression levels of the epithelial markers E-cadherin and occludin. Both E-cadherin (CDH) and occludin (OCLN) were suppressed in Slug-overexpressing cells, and their expression was further suppressed in Daxx-knockdown cells (Fig. 4a; Supplementary Fig. 7a). Conversely, cell migratory and invasive abilities were increased in Slug-overexpressing cells, and were further enhanced by siRNA-mediated knockdown of Daxx (Fig. 4b). In addition, knockdown of Slug in Daxx–silenced CL1–2 and CL–141 cells counteracted Daxx-mediated E-cadherin and occludin downregulation (Fig. 4c; Supplementary Fig. 7b). Increased migration and invasion activity caused by silencing of Daxx was also eliminated by co-silencing Slug and Daxx (Fig. 4d). These results suggest that Daxx suppresses Slug-mediated cell migration and invasion.

We further investigated whether Daxx suppresses cancer cell dissemination and invasion by blocking Slug function. E-cadherin and occludin mRNA levels were downregulated in Slug-over-expressing cells, an effect that was reversed by additional expression of full-length Daxx, but not the DD1 mutant (Fig. 4e; Supplementary Fig. 7c). The consequences of Daxx-mediated regulation of E-cadherin and occludin mRNA was also reflected in the levels of their corresponding proteins, further indicating that Daxx endows the ability to maintain cellular integrity through modulation of epithelial markers (Supplementary Fig. 7d). Similar to the effects of Daxx on E-cadherin, and occludin mRNA and protein levels, full-length Daxx, but not DD1, repressed Slug-induced increases in motility and invasiveness (Fig. 4f). These results suggest that the interaction with Slug is required for Daxx to function as an EMT and invasion suppressor.

**Daxx inhibits cancer metastasis *in vivo*.** To determine whether Daxx is involved in Slug-mediated cancer metastasis *in vivo*, we investigated lung nodule formation in mice intravenously injected with CL1–5 cells stably expressing vector alone, Slug or Slug plus Daxx into NOD–SCID (non-obese diabetic–severe combined immunodeficiency) mice. Mice lungs were collected 35 days after injection, and lung nodules were counted. As shown in Fig. 5a,b, mice injected with CL1–5/Slug displayed more nodules than the control group. In addition, CL1–5/Slug cells co-expressing Daxx displayed a lower average number of lung nodules compared with the CL1–5/Slug group (Fig. 5a,b), suggesting that Daxx may repress a late, Slug-dependent step in cancer metastasis and lung colonization *in vivo*.

To further verify that Daxx acts in local lung metastasis, we investigated lung cancer metastasis in an orthotopic implantation model. CL1–5 cells stably expressing luciferase were infected with viruses containing vector alone, Slug or Slug plus Daxx, and were orthotopically injected into the upper region of the left lung of nude mice. Using an *in vivo* imaging system to detect tumour margin formation in real time, we found that mice in the control group (Neo) showed one strong luciferase signal in the left lung 17 days after implantation, whereas Slug-overexpressing mice showed two or three signals (Fig. 5c). Lungs were collected 25 days after implantation and lung nodules were counted in whole-lung sections. The number of metastatic lung nodules was significantly increased in the Slug-overexpressing group compared with the Neo control group, but was decreased in the Daxx co-expressing group (Fig. 5d,e).

Conversely, loss-of-function studies using CL1–2 cells, in which endogenous Daxx or both Daxx and Slug were knocked down showed that, whereas mice in the control group (shCtrl) did not form nodules in orthotopic lung tumour assays, 44% of mice (four of nine) in the Daxx-knockdown group (shDaxx) formed primary lung tumours and 50% of these primary tumours (two of four) formed metastatic lung nodules (Fig. 5f; Supplementary Fig. 8a,b). Notably, CL1–2 cells with double knockdown of Daxx and Slug failed to form tumours (Fig. 5f). Using these groups of CL1–2 cells for intravenous injection in mice also showed that Daxx-knockdown group formed more metastatic lung nodules compared to the control and the Daxx and Slug double-knockdown groups (Supplementary Fig. 8c). These results indicate that Daxx suppresses lung cancer metastasis dependent on Slug function.

**Daxx[+] correlates with better overall survival in NSCLC.** Previous reports have shown that Slug expression is highly associated with poor overall survival in NSCLC[14,18,41]. Indeed, we found that overall survival was poor in most NSCLC patients with Slug-expressing tumours ($P < 0.001$; Fig. 6a; Supplementary Tables 1 and 2). We further assessed whether Daxx expression

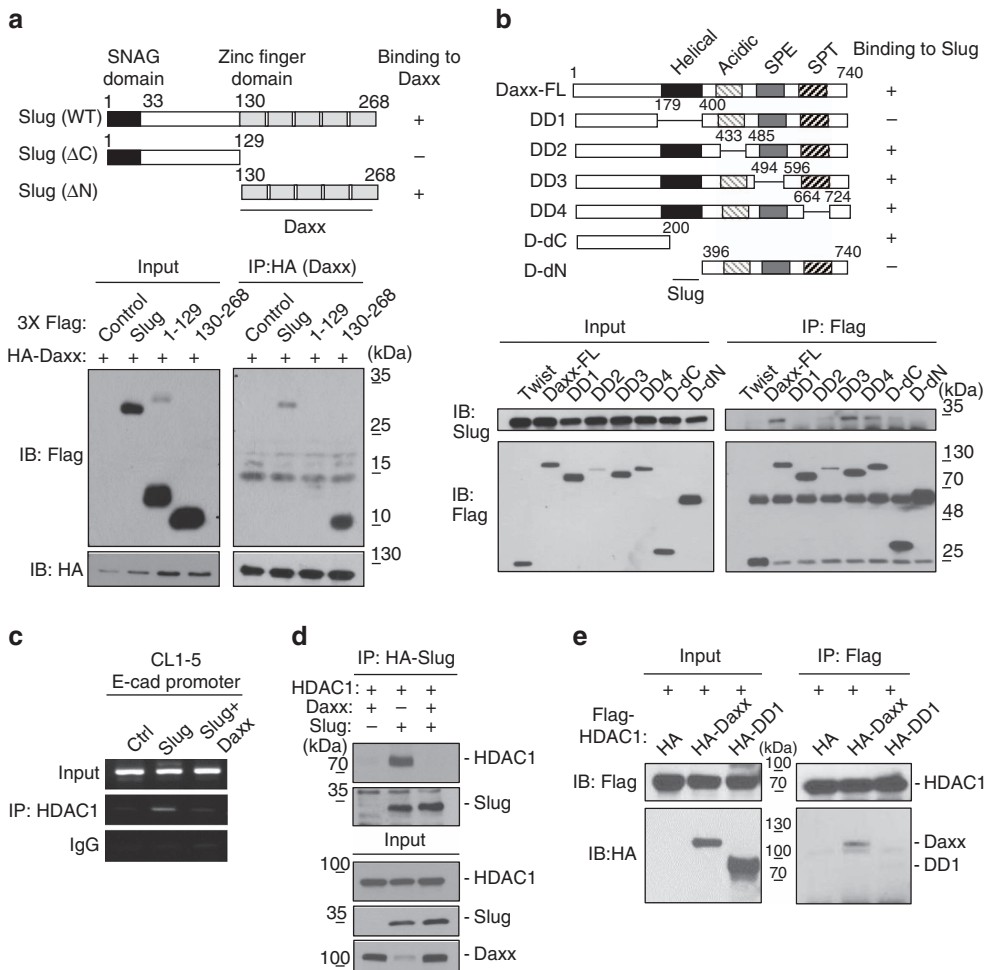

**Figure 3 | Daxx interacts through its helical domain with Slug and HDAC1.** (**a**) Mapping the Daxx-binding domain of Slug. Upper panel: schematic diagram of Slug and Slug deletion mutants. Lower panel: lysates from H1299 cells co-transfected with HA-Daxx and different Flag-Slug constructs or an empty vector for 24 h were co-immunoprecipitated (co-IP) with anti-HA antibodies and fractionated by SDS–polyacrylamide gel electrophoresis (PAGE). Immunoblots were probed with anti-HA or anti-Flag antibodies. (**b**) Mapping the Slug-binding domain of Daxx. Upper panel: schematic diagram of Daxx and Daxx deletion mutants. SPE, Ser/Pro/Glu-rich domain; SPT, Ser/Pro/Thr-rich domain. Lower panel: lysates from H1299 cells co-transfected with HA-Slug and different Flag-Daxx constructs or Flag-Twist for 24 h were immunoprecipitated with anti-Flag antibody and fractionated by SDS–PAGE. Immunoblots were probed with anti-Slug or anti-Flag antibodies. (**c**) ChIP–PCR analyses of HDAC1 on the E-cadherin promoter in CL1–5 cells. CL1–5 cells were transfected with control vector (Ctrl), Slug or both Slug and Daxx plasmids for 30 h, and then ChIP assays were performed using anti-HDAC1 antibody. The precipitated DNA fragments were analysed by PCR to detect the E-cadherin promoter region. The Input and IgG control are same as Fig. 2f which came from identical experiment. (**d**) The effect of Daxx on Slug–HDAC1 interactions was assayed by co-IP. H1299 cells were co-transfected with the indicated plasmids for 30 h. Lysates were immunoprecipitated with anti-HA antibody and fractionated by SDS–PAGE. Immunoblots were probed with the indicated antibodies. (**e**) Daxx binding to HDAC1 through its helical domain was assayed by co-IP. H1299 cells were co-transfected with the indicated plasmids for 30 h. Lysates were immunoprecipitated with anti-Flag antibody and fractionated by SDS–PAGE. Immunoblots were probed with anti-Flag or anti-HA antibodies.

is associated with overall survival in the high-Slug-expressing NSCLC cohort. Using immunohistochemistry, we detected Daxx expression in 83 lung cancer specimens that were positive for Slug expression (Fig. 6b and clinical characteristics is shown in Supplementary Table 3). Kaplan–Meier analyses showed that low Daxx expression was associated with poor overall survival of NSCLC patients whose tumours displayed positive Slug expression ($P < 0.005$; Fig. 6c). In addition, multivariate Cox proportional hazard regression analyses using a stepwise selection model indicated that Daxx expression was an independent predictor of the overall survival of NSCLC patients (hazard ratio (HR) = 0.202, 95% confidence interval (CI) = 0.06–0.681; $P = 0.0099$; Supplementary Table 4). In addition, by dividing the Slug$^+$ cohort into low and high groups, and combining them with the status of Daxx expression, we found that patients in the Daxx$^-$ and Slug$^{high}$ group showed the worst overall survival

($P = 0.001$; Fig. 6d). A further investigation of the Slug downstream target, E-cadherin, in serial sections from the 83-sample NSCLC cohort showed that higher E-cadherin expression correlated with a better survival rate ($P = 0.037$; Fig. 6e). By combining Daxx and E-cadherin expression data, we found that patients negative for both Daxx and E-cadherin expression showed poor overall survival, whereas patients in the Daxx and E-cadherin high-expression group showed a higher survival rate ($P < 0.001$; Fig. 6f). These findings suggest that higher Daxx and E-cadherin expression could serve a protective function (HR = 0.20, 95% CI = 0.005–0.86; $P = 0.0031$; Supplementary Table 4).

Furthermore, a tree diagram to illustrate the preferential expression profile and the conditional probabilities of Daxx, Slug and E-cadherin expressions is shown in supplementary Fig. 9b. The results suggest that Daxx-Slug-E-cadherin pathway does exist

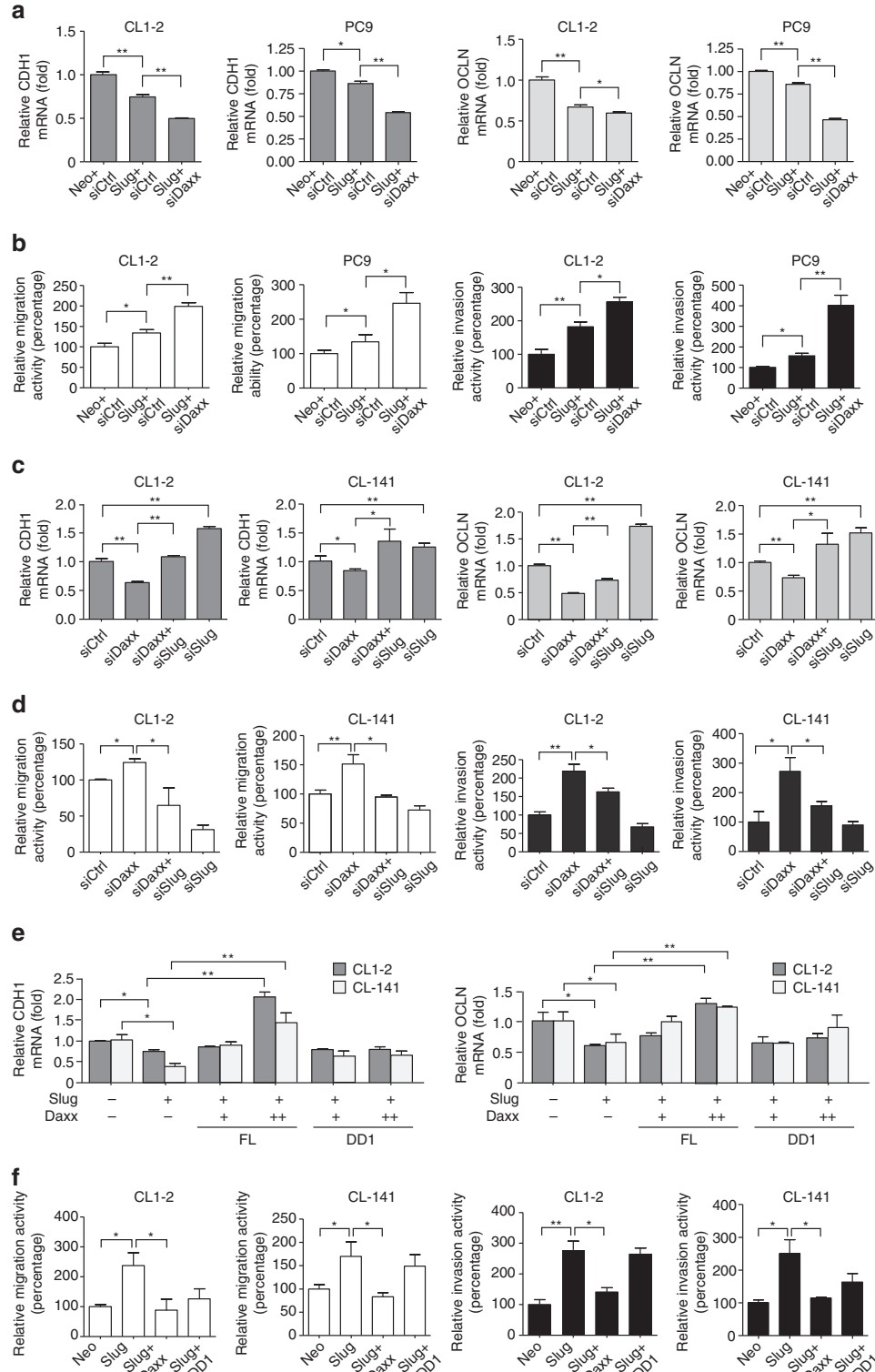

**Figure 4 | Daxx suppresses EMT and cell invasion abilities by restraining Slug activity.** (**a,b**) CL1–2 and PC9 cells stably expressing Neo vector or Slug protein were transfected with control or siDaxx siRNAs. (**a**) Cells were incubated for 72 h, and total RNA was isolated for quantification of E-cadherin (CDH1) and occludin (OCLN) mRNA levels by RT–qPCR. Each target mRNA data point was first normalized to mRNA levels of GAPDH, used as an internal control, and then expressed relative to the control group. (**b**) Cells treated as indicated were used in migration (left) and invasion (right) assays. (**c,d**) Double knockdown of endogenous Daxx and/or Slug in CL1–2 and CL–141 cells. Cells were subsequently analysed for E-cadherin and occludin mRNA expression by RT–qPCR (**c**) and assayed for migration/invasion ability (**d**). Error bars represent s.d. (n = 3). (**e,f**) CL1–2 and CL–141 cells stably expressing control (Neo), Slug, Slug + full-length Daxx (FL) or Slug + Daxx D1 mutation (DD1) were prepared by infection with virus at a multiplicity of infection (MOI) of 1 or 3. mRNA levels of E-cadherin (CDH1) and occludin (OCLN) (**e**) were analysed by RT–qPCR using GAPDH as an internal control for normalization. (**f**) CL1–2 and CL–141 cells stably expressing the indicated proteins were applied for migration and invasion assays. Data are presented as relative fold change compared with Neo controls. Each data point represents the mean ± s.d., n = 3. *P < 0.05, **P < 0.01, paired two-way Student's t-test.

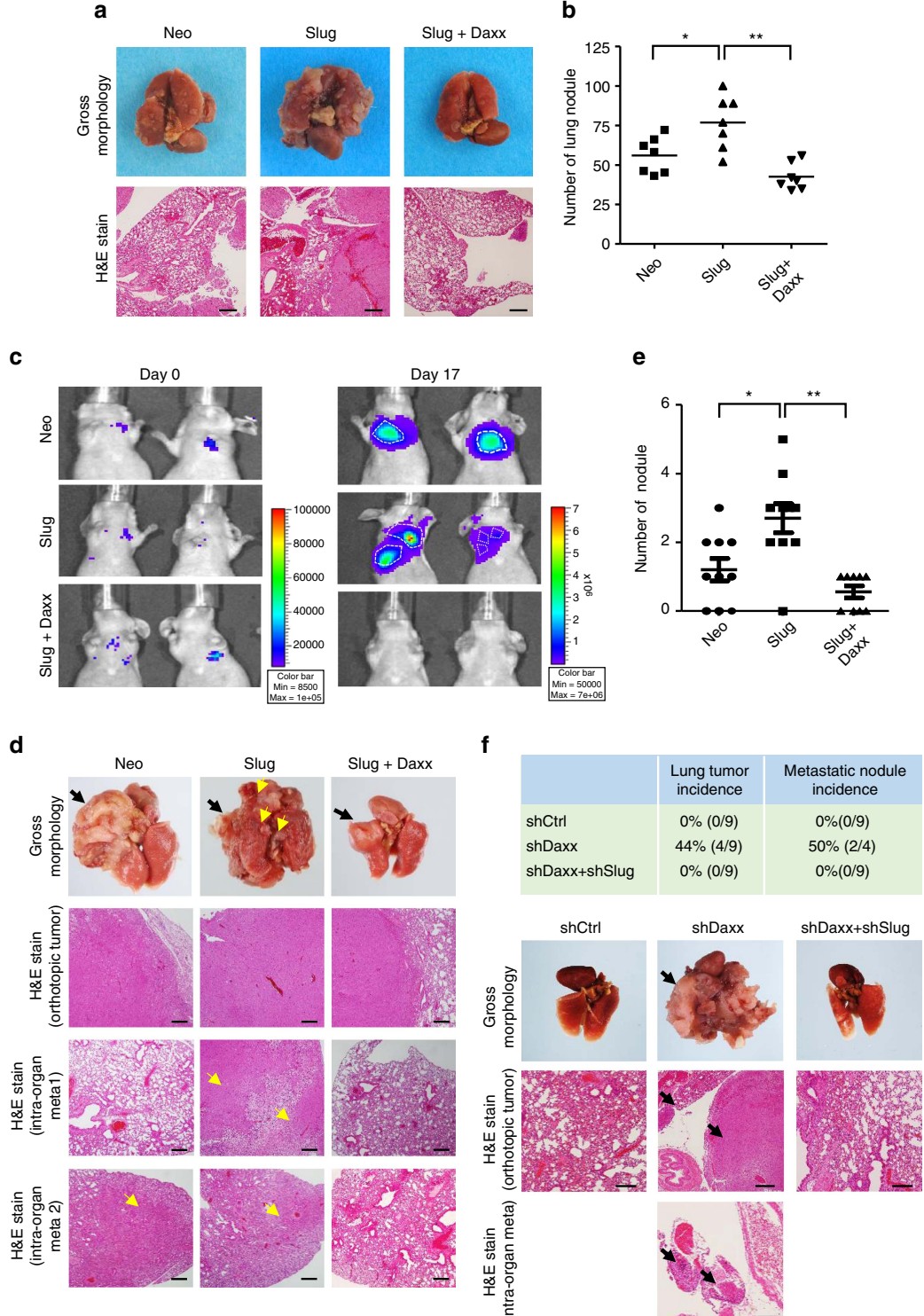

**Figure 5 | Daxx decreases Slug-induced *in vivo* cancer metastasis in tail vein and orthotopic lung cancer transplantation models. (a,b)** Tail vein lung metastasis experiments. CL1-5/Neo, CL1-5/Slug or CL1-5/Slug + Daxx cells were intravenously injected into mouse tail vein to mimic extravasation of lung metastasis. **(a)** Lungs of mice were collected on day 25 after injection, and histologically examined by hematoxylin–eosin (H&E) staining and bright field microscopy. Scale bar, 50 μm. **(b)** Quantitative evaluation of lung metastatic nodules. Data are expressed as means ± s.e.m. (*n* = 7 mice per group). **(c–e)** *In vivo* orthotopic implantation assay. CL1-5 cells stably expressing luciferase were infected with Neo control, Slug or Slug + Daxx lentiviruses, and cells (1 × 10⁵) orthotopically injected into the left lung of nude mice. **(c)** The luciferase signal was detected using an *in vivo* imaging system system at days 0 and 17. **(d)** Twenty-five days after transplantation, mouse lungs were collected, fixed with formaldehyde, sectioned and examined by H&E staining. Scale bar, 50 μm. Arrows indicate the location of tumor. **(e)** Quantitative evaluation of lung metastatic nodules. Data are expressed as means ± s.e.m. (*n* = 10 mice per group). **(f)** *In vivo* orthotopic lung tumour implantation. CL1-2 cells (1 × 10⁵) stably expressing control, shDaxx or shDaxx + shSlug lentiviruses were orthotopically injected into the left lung of nude mice. Mouse lungs were collected 40 days after implantation. Tumour-formation incidence is shown in the upper table. Lungs were fixed and examined by H&E staining. Arrows indicate the location of tumor. *$P < 0.05$, **$P < 0.01$, paired two-way Student's *t*-test.

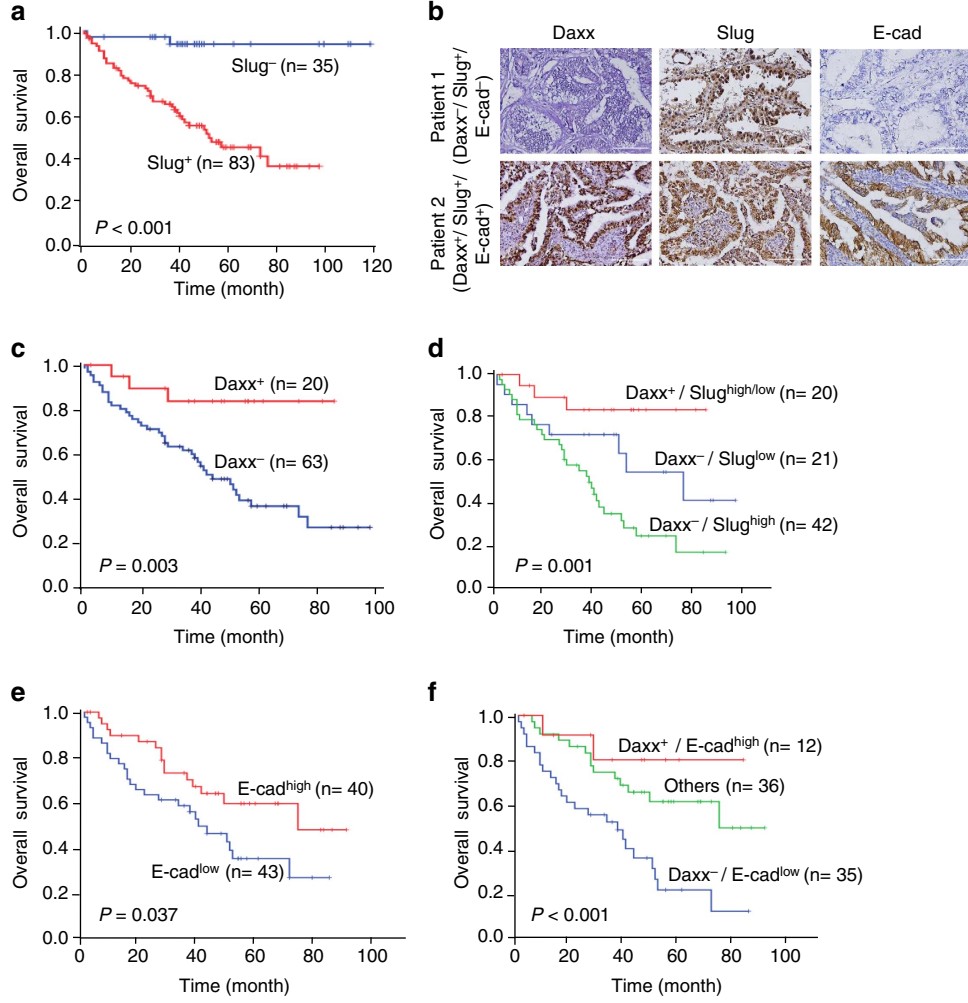

**Figure 6 | Daxx is a prognostic marker for NSCLC patients with Slug-expressing tumors.** (**a**) Kaplan–Meier analysis of overall survival for 118 NSCLC patients, divided into Slug$^-$ and Slug$^+$ groups according to their Slug protein abundance. (**b**) Immunohistochemistry of Daxx, Slug and E-cadherin in serial sections of NSCLC tumour specimens. (**c**–**f**) Kaplan–Meier analysis of overall survival for 83 NSCLC patients with Daxx$^+$ versus Daxx$^-$ (**c**); Daxx versus Daxx$^-$/Slug$^{low}$ and Daxx$^-$/Slug$^{high}$ (**d**); E-cadherin (E-cad) high versus E-cad low(**e**); or Daxx$^+$/E-cad$^{high}$, Daxx$^+$/E-cad$^{low}$ or Daxx$^-$/E-cad$^{low}$ (Others), versus Daxx$^-$/E-cad$^{low}$. (**f**) P-values were obtained from two side log rank tests.

in NSCLC patient and Daxx may serve as a prognostic marker to further distinguish Slug-expressing NSCLC tumours.

**Daxx involves in HIF-1α-mediated invasion and EMT.** Hypoxic tumour microenvironments serve as a major driving force behind cancer metastasis through activation of EMT-TFs and HIF-induced pathways[42]. Therefore, we determined whether the hypoxic microenvironment affected Daxx expression in lung cancer cells. The oxygen-responsive factor, HIF-1α and HIF-2α, were detected in CL1–2 and PC9 cells following exposure to hypoxic conditions (2% $O_2$) for up to 24 h (Fig. 7a). Under hypoxic conditions, Daxx expression was downregulated compared with cells cultured in normal oxygen (10% $O_2$), whereas Slug was upregulated (Fig. 7a). In addition, treatment of cells with different concentrations of the HIFs-α activator, cobalt chloride ($CoCl_2$), caused a concentration-dependent decrease in Daxx expression (Fig. 7b). The downregulation of Daxx mRNA expression was abolished by siRNA-mediated HIF-1α knockdown under both hypoxic and $CoCl_2$-treatment conditions (Fig. 7c,d and Supplementary Fig. 10a), suggesting that upregulation of HIF-1α under hypoxic conditions suppresses Daxx mRNA expression. Although HIF-1α has been shown to upregulate Slug expression during EMT in head and neck squamous cell

carcinoma cells[43], we did not detect changes in Slug mRNA levels in lung cancer cells upon HIF-1α knockdown or exposure to hypoxia (Supplementary Fig. 10b,c), suggesting that alternative pathways are involved in hypoxia-mediated Slug upregulation at the protein level. In addition, our finding indicated that Daxx gene expression in hypoxic lung cancer cells is suppressed via a novel HIF-1α-mediated mechanism.

To further determine whether Daxx is involved in hypoxia-mediated metastatic activity, we investigated lung cancer cell invasion ability and expression of EMT-related genes under hypoxic culture conditions. Hypoxia increased invasive ability in both CL1–2 and PC9 cells compared with normoxia (Supplementary Fig. 11a), and also caused downregulation of E-cadherin and occludin mRNA (Supplementary Fig. 11b,c). To determine whether Daxx-induced increases in EMT and cell invasiveness caused by hypoxia were dependent on HIF-1α, we knocked down HIF-1α using an siRNA approach. The suppression of E-cadherin and occludin mRNAs under hypoxic conditions was reversed by HIF-1α knockdown (Fig. 7e; Supplementary Fig. 11b), suggesting the HIF-1α negatively regulates EMT markers in hypoxia. Silencing HIF-1α also abolished the hypoxia-induced increase in cell invasiveness, whereas double knockdown of Daxx and HIF-1α restored normal

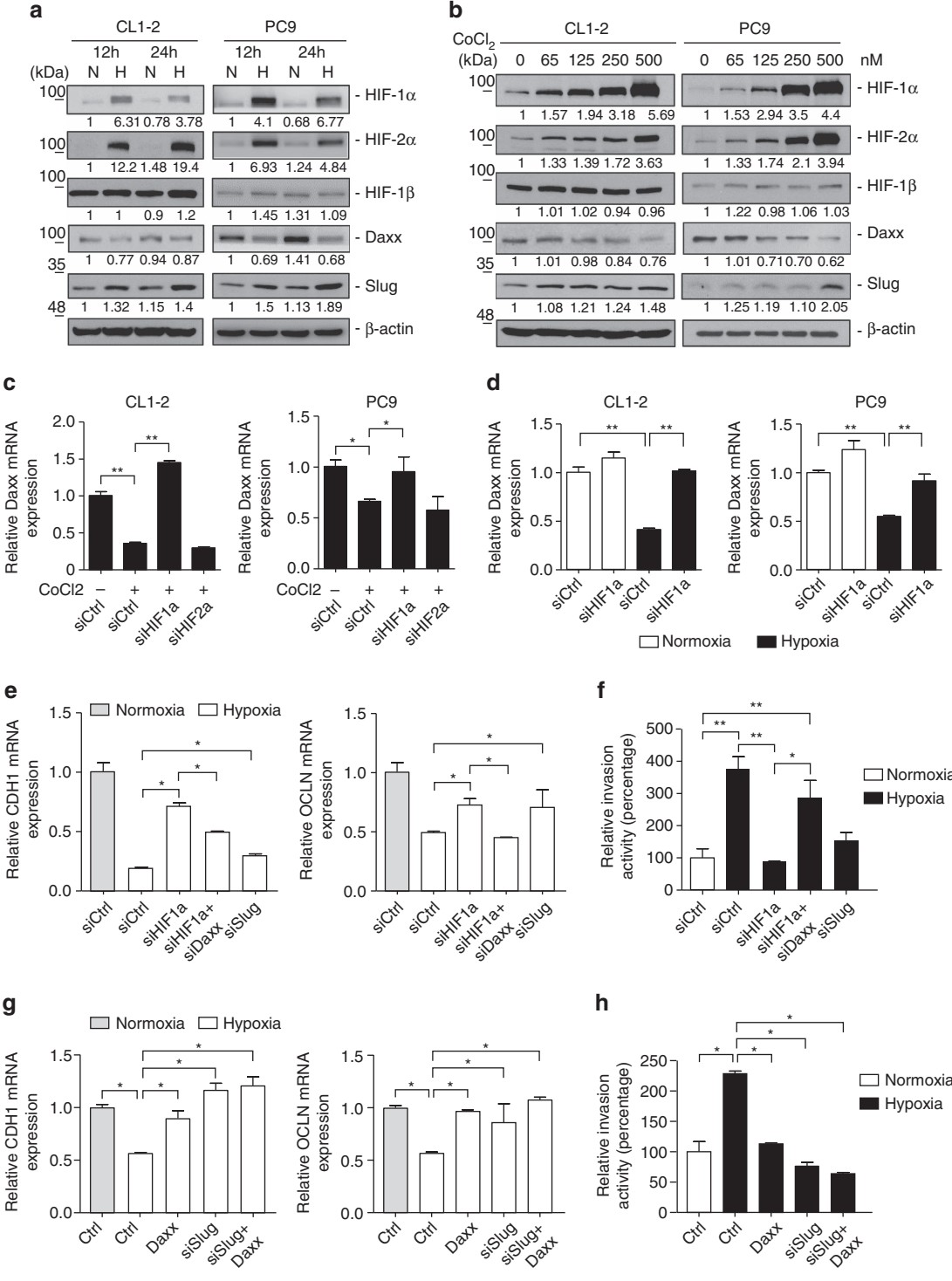

**Figure 7 | Daxx is involved in hypoxia-induced EMT and cell invasion.** (**a**) CL1–2 and PC9 cells were incubated under normoxic (N) and hypoxic (H) conditions for 12 and 24 h, and expression of the indicated proteins was detected by immunoblotting. (**b**) CL1–2 and PC9 cells were treated with increasing concentrations of the HIF-α inducer CoCl2 for 24 h. Cell lysates were then collected, and the indicated proteins were detected by immunoblotting. (**c**) CL1–2 and PC9 cells were transfected with control siRNA or siRNAs targeting endogenous HIF-1α or HIF-2α for 48 h and then incubated with or without 125 nM CoCl$_2$. Daxx mRNA expression was quantified by RT–qPCR. (**d**) CL1–2 and PC9 cells were transfected with control siRNA (siCtrl) or siHIF-1α and subsequently incubated under normoxic or hypoxic conditions for 48 h. Total RNA was isolated for quantification of Daxx mRNA expression by RT–qPCR. (**e,f**) CL1–2 cells transfected with control siRNA (siCtrl), shHIF-1α, siHIF-1α + siDaxx or siSlug were subjected to transwell under normoxic and hypoxic conditions. (**g,h**) CL1–2 cells infected with control vector (Ctrl), Daxx, siSlug or siSlug + Daxx were subjected for RT–qPCR for CDH1 and OCLN mRNA detection (**g**) or transwell invasion assay (**h**) under normoxic or hypoxic conditions. For quantitative PCR assays, levels of target mRNAs were normalized to those for GAPDH and expressed relative to the control group. All data are presented as the relative ratio ± s.d. of triplicate determinations. *$P < 0.05$, **$P < 0.01$, paired two-way Student's $t$-test.

invasiveness and CDH/OCLN expressions (Fig. 7e,f). Notably, re-expression of Daxx attenuated the hypoxia caused down-regulation of E-cadherin and occludin expression in CL1–2 and PC9 cells (Fig. 7g; Supplementary Fig. 12a,b), as evidenced by the diminished invasiveness of Daxx-overexpressing cells during hypoxia (Fig. 7h; Supplementary Fig. 12c). Moreover, silencing Slug abolished the hypoxia-induced decrease in CDH/OCLN expressions and increase in cell invasiveness, whereas Daxx re-expression along with Slug silencing had similar effects as Slug silencing alone (Fig. 7g,h). These results suggest that HIF-1α-mediated Daxx downregulation contributes to hypoxia-induced cell dissemination, invasion and metastasis of lung cancer cells by preventing Daxx from blocking Slug activation.

**HIF-1/Daxx axis predicts outcome of lung cancer patients.** Finally, to evaluate whether the HIF-1α and Daxx regulatory axis is associated with the overall survival rate in lung cancer patients, we analysed the GSE31210 data set, downloaded from the gene expression omnibus (GEO) database[44]. This data set contains mRNA expression profiles of 226 lung adenocarcinoma patients. The patients were divided into three groups according to the mRNA expression level of Daxx and HIF-1α-responsive genes; these latter genes were collected from a study that identified HIF-targeted genes as core responders to hypoxia[45]. We found that, under hypoxic conditions (HIF-1α-responsive gene—high), low levels of Daxx expression were significantly associated with poor overall survival ($P < 0.0001$; Fig. 8a). Consistent with these finding, multivariate Cox proportional hazard regression analyses using a stepwise selection model indicated that Daxx expression under hypoxia is a protective factor (HR = 0.48, 95% CI = 0.3–0.75; $P = 0.001$; Supplementary Table 5). These findings

show that the expression of Daxx is involved in hypoxia-induced lung cancer progression.

## Discussion

In this study, we determined the mechanism underlying the ability of Daxx to suppress hypoxia-induced lung cancer metastasis, showing that Daxx acts through the HIF-1α/HDAC1/Slug pathway (Fig. 8b). Specifically, we showed that Daxx suppresses cancer metastasis by interfering with Slug E-box binding and competing for recruitment of the effector HDAC1 by directly binding to the Slug DNA-binding domain. The dysregulation of Daxx/Slug regulatory axis in lung cancer plays distinct role with EGFR, KRAS, HER2 and BRAF mutations status, as these genetic mutations have been found to highly associate with the progression of lung adenocarcinoma[46] (Supplementary Table 6). Most importantly, we revealed a novel mechanism whereby Daxx is downregulated by HIF-1α, thus enabling initiation of metastasis in cancer cells. Consistent with this, Daxx overexpression reversed hypoxia-induced tumour dissemination and invasiveness in lung cancer cells. We also clinically confirmed that high Daxx expression under hypoxia predict better overall survival in NSCLC patients. Therefore, Daxx may be a potential therapeutic target for strategies designed to inhibit cancer metastasis driven by hypoxia and the HIF-1α pathway.

Accumulating evidence has shown that Daxx is a multi-functional protein. For example, Daxx is overexpressed in ovarian cancer tissues and promotes the development of ovarian tumours[47]. However, Daxx has been found to act as a tumour suppressor by repressing several oncogenes in other types of cancer, including prostate cancer, colon cancer and breast cancer[37,48–50]. These findings imply that the distinct cellular contexts and gene expression profiles of different types of cancers

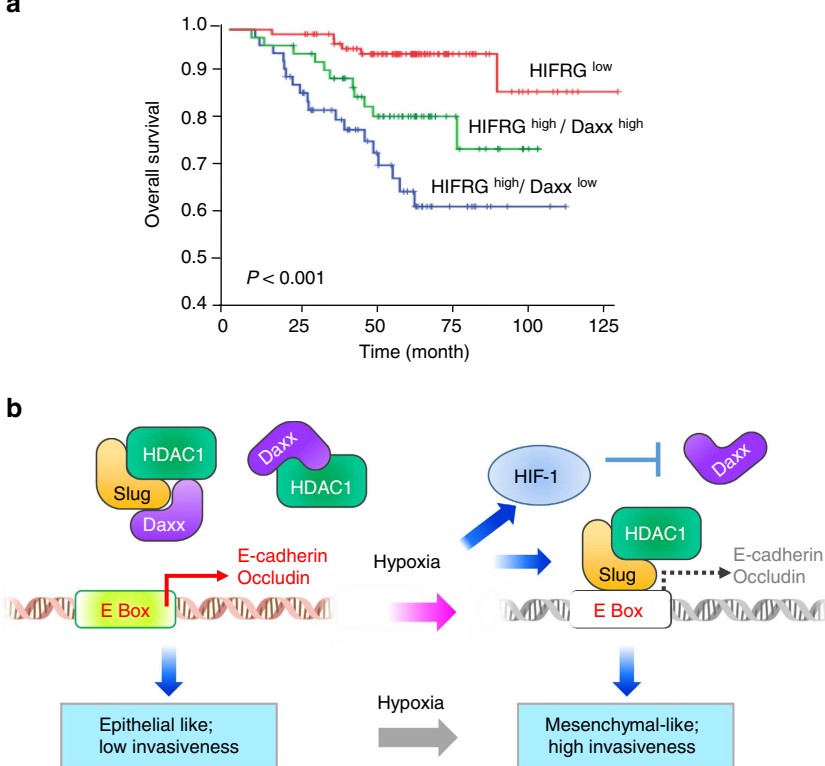

**Figure 8 | Hypoxia-induced Daxx mRNA expression is associated with increased overall survival of lung adenocarcinoma. (a)** Kaplan–Meier analysis of overall survival for 226 lung adenocarcinomas based on expression profiles of Daxx and HIF-1-responsive genes (HIFRGs). The patients were divided into three groups—HIF[low], HIF[high]/Daxx[low] and HIF[high]/Daxx[high]—according to the abundance of Daxx and HIFRGs. **(b)** Model of Daxx inhibition of hypoxia-induced EMT through suppression of the HIF-1α/Slug/E-cadherin pathway.

determine the role of Daxx. As a major metastatic driver in lung cancer, over-activated Slug promotes cell invasive ability, which usually results in cancer recurrence and poor clinical outcomes[41]. Our results demonstrated that Daxx expression is inversely correlated with cell invasiveness and patient survival, as well as with the ability to block Slug function, thus revealing a novel role for Daxx as a metastatic suppressor in lung cancer.

In addition to promoting metastasis, overexpression of Slug could also provide a survival signal through suppression of the DNA-damage sensor, Puma (BBC3)[51,52], suggesting that suppressing Slug function could be a potential therapeutic approach for the treatment of cancer. We found that the helical domain of Daxx interferes with Slug-mediate dissemination and invasion through direct interactions, indicating that the Daxx helical domain potentially acts as a Slug suppressor. Interestingly, co-immunoprecipitation assays showed that Daxx also binds to Snail and another EMT-TF, Twist (Supplementary Fig. 13). Because Daxx directly interacts with Slug DNA-binding motifs, which share a similar C2H2 motif with Snail and ZEB1/2 (ref. 11), Daxx could also act as a DNA-binding blocker for other EMT-TFs that contain zinc fingers. These results also imply that Daxx tightly controls the EMT process through EMT-TFs. However, whether Daxx downregulates Snail/Twist transcriptional activity remains to be investigated.

Recent work has shown that malignant cancer cells can activate several oncogenic pathways in hypoxic environments, enabling them to escape the threat of an oxygen shortage[53]. Components of these pathways include the oxygen-sensitive HIFs, which activate metastasis cascades by endowing cancer cells with invasive and EMT properties, stimulating cells to migrate to nutrient-rich areas in distant organs[54,55]. We found that in lung cancer, hypoxic microenvironments not only upregulated Slug but also downregulated Daxx.

We also found that the downregulation of Daxx under hypoxia was regulated at the transcriptional level by HIF-1α. Although we did not find a hypoxic response element within the Daxx promoter, we cannot exclude the possibility that HIF-1α suppresses Daxx mRNA at transcriptional or post-transcriptional levels, because HIF-1α has been found to regulate EPO (erythropoietin) through its 3′-untranslated region and IGFBP1 (insulin-like growth factor-binding protein 1) through its first intron[56]. Thus, determining how HIF-1α downregulates Daxx expression is important in uncovering the mechanism of hypoxia-induced cancer metastasis.

In addition to the physical role of Daxx in repressing Slug-mediated cell invasiveness, we also investigated Slug-related events after Daxx binding. Daxx acts as a scaffolding protein and forms protein complexes with other kinases or epigenetic modifiers, through which it facilitates its own down-stream actions[57–59]. For example, the tethering of Daxx and HIPK2 (homeodomain-interacting protein kinase 2) on axin enhances HIPK2-mediated phosphorylation of serine-46 in p53, increasing the transcriptional activity of p53 (ref. 60). Moreover, the Daxx and ATRX complex is required for histone H3.3 heterochromatin deposition and accounts for H3.3 serine-31 phosphorylation; these mechanisms are abolished by disrupting the Daxx–ATRX complex[59,61]. Recent findings have shown that the Slug protein is tightly regulated at the posttranslational level and its transcriptional activity is negatively controlled by several kinases via a phosphodegron mechanism[41,62,63]. To address this, we evaluated whether Daxx-bound Slug is subsequently targeted for degradation. However, an analysis showed that the half-life of Slug protein was unaltered in the absence or presence of Daxx. We thus conclude that Daxx binding might not affect Slug protein stability or that it may require the presence of another, as yet unidentified, protein.

In summary, we demonstrated that Daxx plays a pivotal role in hypoxia-induced cancer cell dissemination and invasion by regulating the HIF-1α/HDAC1/Slug and E-cadherin pathway in lung cancer. Accordingly, Daxx may be a potential therapeutic target for inhibition of cancer metastasis.

## Methods

**Cell culture.** The human lung cancer cell lines PC9, H23, HOP-62, HOP–92 and H460 were purchased from the Developmental Therapeutics Program of the National Cancer Institute (NCI, Bethesda, MD, USA). H1975 and PC9 cell lines were kindly provided by Professor Chi-Hsin Yang (National Taiwan University, Taipei). These cell lines are maintained in RPMI medium containing 10% fetal bovine serum. H1299, HeLa and HEK-293 cells (purchased from the American Type Culture Collection) were cultured in Dulbecco's Modified Eagle Medium containing 10% fetal bovine serum. The CL1–2 and CL1–5 lines were derived from the CL1-0 cell line by in vitro Transwell selection, as described previously[64]. CL–141 and CL100 lines, derived from clinical patients, were established in our laboratory and have been authenticated by karyotyping and STR analysis. We tested the listed cell lines for mycoplasma contamination using a polymerase chain reaction (PCR)-based enzyme-linked immunosorbent assay (ELISA; Sigma). Stably transduced pooled clones were maintained in medium supplemented with $0.75 \mu g \, ml^{-1}$ puromycin or $400 \mu g \, ml^{-1}$ G418 as required for the corresponding selection markers. For hypoxic culture, the incubator (ASTEC, Fukuoka, Japan) atmosphere contained 5% $CO_2$, 2% $O_2$ and 93% $N_2$, whereas the normoxic atmosphere contained 95% air and 5% $CO_2$.

**Viruses and transduction.** Daxx, Daxx-DD1 and Slug were subcloned into the pAS2.neo lentiviral vector (obtained from the National RNAi Core Facility in Academia Sinica, Taipei, Taiwan) and prepared in accordance with standard protocols. Briefly, HEK293T cells were co-transfected with pLKO.1-shRNA, pCMVΔR8.91 and pMD.G, and virus-containing medium was collected at 24, 48 and 72 h post transfection. Stably transduced clones were generated by first infecting cells with lentivirus in medium containing polybrene ($8 \mu g \, ml^{-1}$), and then selecting a pool of antibiotic-resistant clones by treating cells for 24 h after infection with $400 \mu g \, ml^{-1}$ G418.

**RNA interference and chemical treatment.** Transient knockdown experiments were conducted using human siRNA-SMARTpool for Daxx, Slug, HIF-1α and siControl (Thermo Scientific, MA, USA). The corresponding sequences are listed in Supplementary Table 7. CL1–2, PC9 or CL141 cells were transfected with the indicated siRNAs using Lipofectamine RNAiMAX transfection reagent (Invitrogen, Taipei, Taiwan) according to the manufacturer's protocols. Invasion and migration assays were performed 48 h post transfection, and protein and mRNA expression levels were determined 72 h post transfection.

**Cell lysate preparation and immunoblotting.** All experiments were performed in accordance with standard protocols. Briefly, cell lysates for immunoblotting were prepared in RIPA buffer (1% Nonidet P-40, 0.5% sodium deoxycholate and 0.1% SDS) containing complete protease inhibitor cocktail (Roche). Protein samples were separated by SDS–polyacrylamide gel electrophoresis, transferred to poly (vinylidene difluoride) membranes and probed with the indicated antibodies. Proteins were detected by enhanced chemiluminescence. Scans of enhanced chemiluminescence (ECL) films showing uncropped blots are presented in Supplementary Fig. 14.

**Antibodies.** Primary antibodies used for immunoblotting were as follows: monoclonal anti-HA (1:5,000; HA11; Covance), anti-Flag (1:5,000; M2; Sigma Aldrich), anti-Daxx (1:2,000, M112; Santa Cruz Biotechnology), anti-E-cadherin (1:1,000; 610182, BD Biosciences), anti-occludin (1:2,000; 13409, Proteintech), anti-vimentin (1:10,000; 550513, BD Biosciences), anti-N-cadherin (1:1,000; 610921, BD Biosciences), polyclonal anti-Slug (1:1,000; G18, Santa Cruz Biotechnology), anti-β-actin (1:10,000; AC-15, Sigma Aldrich), anti-HIF-1α (1:2,000; 610958, BD Biosciences), anti-HIF-2α (1:1,000; 7096, Cell Signaling) and anti-HIF-1β (1:2,000; 5537, Cell Signaling).

**Immunoprecipitation.** For co-immunoprecipitation, H1299 cells were transfected with plasmids expressing Daxx and Slug with the indicated tags for 24 h and then treated with the proteasome inhibitor MG132 (10 μM) for 5 h. Cells were lysed in IP lysis buffer (20 mM Tris pH 7.5, 100 mM NaCl, 1% Nonidet P-40, 100 mM Na3VO4, 50 mM NaF and 30 mM sodium pyrophosphate) containing complete protease inhibitor cocktail (EDTA-free; Roche Applied Science, Mannheim, Germany), and lysates were cleared by centrifugation. The resulting supernatant was incubated with anti-Flag M2 antibodies overnight at 4 °C and with protein G agarose for 1 h at 4 °C. For endogenous co-immunoprecipitation, the cells were treated with MG132 (10 μM) for 5 h, and Slug or Daxx was immunoprecipitated using the corresponding antibodies.

**Luciferase reporter assay.** H1299 cells were seeded into six-well plates ($3 \times 10^5$ cells per well) and transfected with plasmids using Lipofectamine 2000 reagent (Invitrogen, Carlsbad, CA, USA). The luciferase reporter construct, $3 \times$ Slug-binding sites luciferase, was co-transfected with the activator construct, pGal4-VP16, at a DNA ratio of 10:1. Another luciferase plasmid containing the wild-type or mutant E-cadherin promoter was also used to detect luciferase activity. Both of these constructs were co-transfected with the pRL-Renilla plasmid to normalize for transfection efficiency. Luciferase and Renilla activity were measured using a Dual Luciferase Reporter Assay System (Promega, WI, USA) at 24 h post transfection. The data were expressed as the relative luciferase activity (firefly luciferase activity divided by Renilla activity).

**EMSA.** The full-length Slug complementary DNA (cDNA) sequence was cloned in frame with the 5′-terminal GST sequence into the pGEX-4T-1 vector. GST-fusion proteins were expressed in *Escherichia coli* strain BL21 (Invitrogen), purified with glutathione Sepharose 4B (GE Healthcare) and used for EMSA. The concentration of purified GST-Slug protein was determined using Coomassie blue staining by reference to a standard curve prepared from a commercial bovine serum albumen protein standard. Full-length Daxx cDNA was cloned in frame with the 5′-terminal HA sequence into the pCDNA3.1 vector. Daxx protein was produced by *in vitro* translation using the TNT Quick Coupled Transcription/Translation System (Promega), according to the manufacturer's specifications. The expression of HA-Daxx was confirmed by immunostaining. Wild-type (GACTTCCGCAAGCT CACAGGTGCTTTGCAGTTCCGACG) and mutant (GACTTCCGCAAGCTCA TAGGTTCTTTGCAGTTCCGACG) E-box-containing oligonucleotides (E-boxC 1.75 pmol μl$^{-1}$) were labelled with [γ-$^{32}$P] ATP using T4 polynucleotide kinase. Unincorporated oligonucleotides were removed using G-25 spin columns (GE- Healthcare), and the relative activity of each probe was determined using a gamma counter. For protein-probe binding, 1 μg of Slug protein, with or without Daxx, was incubated with 0.05 nM $^{32}$P-labelled E-boxC in binding buffer at room temperature for 30 min. The $5 \times$ binding buffer contained 2.5 mM EDTA, 50 mM Tris-HCl (pH 7.5), 2.5 mM DTT, 5 mM MgCl2, 250 mM NaCl, 20% (w/v) glycerol and 0.25 mg ml$^{-1}$ poly(dI-dC)-poly(dI-dC). The binding reaction mixture was run on a 4% non-denaturing polyacrylamide gel at 350 V for 3 h in $0.5 \times$ TBE buffer. The gel then was placed on filter paper and dried in a gel dryer. The isotope signal was determined by exposure to X-ray film.

**Chromatin immunoprecipitation.** CL1–5 cells ($1 \times 10^7$) stably expressing Neo vector, Slug or Slug + Daxx were analysed using Magna ChIP A/G (Millipore, MA, USA) according to the manufacturer's instructions. Briefly, cellular DNA was cross-linked in culture media containing 1% formaldehyde and quenched by addition of 0.125 M glycine. After washing with cold PBS, the cells were removed by scraping, and soluble chromatin lysates were extracted by sonication and centrifugation. Cleared lysates were then mixed with anti-Slug (G18, Santa Cruz Biotechnology), anti-HDAC1 (17-10199, Millipore) or anti-Daxx (M112, Santa Cruz) antibodies and protein A/G magnetic beads overnight at 4 °C. The complexes were pelleted and washed with the indicated buffers. Formaldehyde cross-links were reversed by eluting the DNA/protein solution with proteinase K-containing elution buffer at 65 °C for 2 h. The DNA eluates were purified, and E-cadherin was amplified by PCR using the primers pair, 5′-CGA ACC CAG TGG AAT CAG AA-′3 (forward) and 5′-GCG GGC TGG AGT CTG AAC TG-3′ (reverse).

**Real-time RT–PCR.** Total RNA was extracted using the TRIzol reagent (Invitrogen), and cDNA was prepared and used for PCR analyses. The amount of PCR product was determined by agarose gel electrophoresis in the presence of ethidium bromide. For quantitative PCR with reverse transcription (RT–qPCR), primer sets for DAXX (Hs00985566_g1), Slug (Hs00161904_m1), E-cadherin (Hs01023894_m1), occludin (HS00170162_m1) and the internal control GAPDH (Hs99999905_m1) were purchased from Applied Biosystems (CA, USA). Data were acquired using an ABI PRISM 7500 system (Applied Biosystems). The expression level of each gene relative to that of GAPDH was defined as $-\Delta CT = -(CT \times X - CT \times GAPDH)$. The gene expression/GAPDH cDNA ratio was calculated as $2 - \Delta CT \times K$, where K is a constant. Experiments were performed in triplicate.

**Invasion and migration assays.** For invasion assays, transwell chambers (8 μm; Falcon, NJ, USA) were coated with Matrigel (30 μg; BD Biosciences, Bedford, MA, USA). For both migration assays and invasion assays, the plates were seeded with $10^5$ cells per well and incubated for 20 h. Cells that invaded or migrated from the top to the bottom of the chamber were fixed in methanol and stained with 50 μg ml$^{-1}$ propidium iodide. The cells were subsequently photographed under a fluorescence microscope and counted using AIS software (Imaging Research Inc., St Catherine's, Canada). Each condition was assayed in triplicate.

3D invasion assays were performed by first pre-coating 96-well culture plates with Matrigel (BD Biosciences) and incubating overnight at 37 °C to allow Matrigel polymerization. Cells ($5 \times 10^3$), labelled with Cyto-IDTM Red (ENZO Life Sciences, Plymouth Meeting, PA, USA), were seeded on Matrigel-coated plates, and cell invasion ability was observed under a Leica DMI6000B fluorescence

microscope (Leica Microsystems GmBH, Wetzlar, Germany) for 24 h. The velocity and motility tracts of cell migration were analysed and quantified using MetaMorph 7.7.5 software.

**Animal model.** Mice were randomly divided into three groups with the same number of animals in each group. In cases of technical failure, such as inaccurate orthotopic or intravenous injection, the corresponding samples were excluded from the final analysis. For intravenous injections, $1 \times 10^6$ tumour cells were suspended in 0.1 ml of PBS and injected into the lateral tail vein of male NOD–SCID mice (NOD.CB17-Prkdcscid/NcrCrlBltw strain, 6 weeks, LASCO, Taiwan; $n = 10$). All mice were sacrificed 6 week post injection, after which lungs were removed and fixed in 10% formalin. The number of lung tumour colonies was counted under a dissecting microscope. Representative lung tumours were embedded in paraffin, sliced into 4-mm sections and stained with hematoxylin–eosin for histological analysis.

For orthotopic tumour implantation assays, lentivirus-infected cells harbouring the CL1–5/Neo vector (controls), overexpressing CL1–5/Slug or CL1–5/Slug/Daxx, or expressing CL1–2/shCtrl, CL1–2/shDaxx or CL1–2/shDaxx + shSlug ($10^5$ cells in 20 μl of PBS containing 10 μg of Matrigel) were injected into the pleural cavity of male 6-week-old BALB/c nude mice (BALB/cAnN.Cg-Foxn1nu/CrlNarl strain, NAR labs, Taiwan; 10 mice per group). Mice were killed by $CO_2$ anaesthesia 28 days after implantation, and their lungs were removed and fixed in 10% formalin. Lung nodules were counted by gross and microscopic examination.

All animal experiments were performed in accordance with the animal guidelines of the Department of Animal Care, Institute of Biomedical Sciences, Academia Sinica, Taipei, Taiwan (Approve No. 2011099).

**Patient sample collection and immunohistochemistry.** A total of 118 patients who underwent surgical resection for NSCLC at the National Taiwan University Hospital between January 1996 and December 2005 were studied. NSCLC tumours were staged in accordance with the American Joint Committee on Cancer Staging and histological analyses were performed according to World Health Organization standards. None of the patients had received preoperative adjuvant chemotherapy or radiation therapy. The immunohistochemical analysis of paraffin-embedded, formalin-fixed surgical specimens was performed using specific antibodies against Slug (ASC10469, Abgent), Daxx (HPA008736, Sigma-Aldrich) and E-cadherin (610182, BD Biosciences); PBS (instead of primary antibodies) was applied as a negative control. Immunohistochemistry results were reviewed and scored independently by two pathologists. This study was approved by the Institutional Review Board and conducted with the informed written consent of all patients.

**Statistical analysis.** Sample sizes were chosen based on pilot experiments that ensured adequate statistical power with similar variances. Data are presented as means ± s.e.m. or means ± s.d. Differences between two groups were analysed with Student's $t$-test for continuous variables or Pearson's $\chi^2$-test for categorical variables. The overall survival for patient groups with different expression signatures was estimated using the Kaplan–Meier method, and two-sided log-rank tests were applied to compare the differences. Statistical analyses were performed using SPSS for Windows (version 10.0; SPSS, Inc.), and a $P < 0.05$ was considered statistically significant. All $n$ values defined in the legends refer to biological replicates. The researchers involved in the study were not completely blinded during sample collection or data analysis.

**Data availability.** The data that support the findings of this study are included in the article and Supplementary Information or available from the corresponding author on request.

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

## Acknowledgements

We thanks Chen-En, Yeh, Ting-Fang, Che and Prof Szu-Hua, Pan (National Taiwan University) for providing technical supports. We thank Dr Kang-Yi, Su (Department of Clinical Laboratory Sciences and Medical Biotechnology, College of Medicine, National Taiwan University), and NCFPB Integrated Core Facility for Functional Genomics and NRPB Pharmacogenomics Lab for MALDI-TOF MS assay support. We acknowledge 118 patients who provide their surgical resections and their agreement for publish. This work was supported by grants from the National Science Council (NSC), Taiwan (NSC102-2321-B-002-053-; NSC102-2325-B-006-016-; and NSC 99-2628-B-006-031-MY3), the Ministry of Science and Technology, Taiwan (MOST 103-2321-B-002 -022; MOST 104-2314-B-002 -228 -MY3), and National Taiwan University (NTU103R7601-2; NTU104R7601-2). Shu-Ping Wang was supported by a Human Frontier Science Program long-term fellowship.

## Author contributions

T.-M.H. and P.-C.Y. directed the project and contributed equally to this work. C.-W.L. performed and analysed most of the experiments. L.-K.W., S.-P.W., Y.-L.C., Y.-Y.W., W.-Y.L., H.-H.L., S.-C.Y., M.-W.L. and C.-Y.C. provided reagents and materials, and performed data analysis. Y.-L.C., H.-Y.C., T.-H.H. and Y.-H.C. collected and analysed samples from lung cancer patients.

## Additional information

**Competing financial interests:** The authors declare no competing financial interests.

