## [Peer Review File · Nature Communications]

Reviewer #1 (Remarks to the Author)

The manuscript entitled "Daxx Inhibits Hypoxia-Induced Lung Cancer Cell Metastasis by Suppressing the HIF-1 α /HDAC1/Slug Axis" by Lin et al. identified Daxx as being critical in preventing EMT and invasion in lung tumor cells. Mechanistically, the study suggests that Daxx binds to Slug and inhibits HDAC1 binding to Slug, thus antagonizing the ability of Slug to bind and suppress the E-cadherin promoter. Furthermore, the authors provided some data suggesting that HIF1 α could repress Daxx expression, and overexpression of Daxx can suppress HIF1 α -induced invasion. In vivo, overexpression of Daxx can suppress metastasis. In patients, the HIF1 α , Daxx and Slug pathway is associated with outcome.

In general, identification of Daxx as a suppressor of EMT and tumor invasion is novel and interesting, thus worth further investigation. The association of Daxx with good outcome is striking and interesting. However, the proposed mechanism linking Daxx and Slug is not well developed. Especially, the physiological significance of endogenous Daxx in suppressing Slug to maintain epithelial identity is not supported by the experimental data.

Major points:

1. The expression level of Daxx and Slug in epithelial vs. mesenchymal lung cancer cells questions whether endogenous Daxx indeed is required to suppress Slug in epithelial lung cancer cells.

Fig. S1a shows that all four epithelial cells express Daxx, but not Slug. Therefore, in these four epithelial tumor cell lines, Daxx is not needed to suppress Slug since Slug is not expressed in these cells. In the three mesenchymal cells presented in Fig. S1a, only CL-141 express both Daxx and Slug, but E-cadherin is completely suppressed, suggesting that Daxx is not able to suppress Slug in CL-141 cells. It is possible that Daxx level is too low in CL-141 cells to suppress Slug. Then comparing the level of Daxx expression in Fig. S1a and Fig. S1b, it is clear that the Daxx level is highest in CL1-0 cells and second highest in CL1-2 and then CL1-5 cells, much higher than H23, H1975 and PC9 cells. But even such high levels of Daxx in CL1-2 and CL1-5 cells could not suppress Slug in these cells to promote E-cadherin expression, thus questioning whether Daxx is truly involved in inhibiting the mesenchymal state. Instead, the E-cadherin status seems to be mostly correlated with

Slug expression level in these three cell lines presented in Fig. 1b.

2. Although data using overexpression of Daxx and Slug support their interaction and the potential role of Daxx in suppressing Slug transcriptional activity, all the data examining

endogenous Daxx and Slug shed doubts on the significance of this interaction. For example, Fig. 1C shows that knockdown of Daxx in PC9 cells could downregulate E-cadherin, but this could not be due to Slug since PC9 cells show no Slug expression (Fig.S1a). For CL1-2 cells, to show that this effect is due to Slug suppression, double knockdown of Slug and Daxx is needed to show that E-cadherin suppression upon Daxx knockdown is due to Slug. Fig. 2B intends to show that endogenous Daxx and Slug interact, but the Co-IP signals are very weak with lots of background, therefore not convincing.

3. Fig. 3, 4, 5 and 7 are largely based on overexpression of Daxx and Slug. Given all the issues discussed above on endogenous Daxx and Slug function, it is unclear whether physiological function of Daxx is indeed what described for overexpressed Daxx. For example, CL1-5 is the most invasive cells discussed in the paper. Instead of overexpressing Slug and Daxx in already invasive CL1-5 cells used in Fig. 5, The more critical experiment is to test whether removal of endogenous Daxx in CL1-2 cells could promote invasion and metastasis in a Slug-dependent manner.

4. Fig. 6 shows that expression of Daxx is correlated with good survival, while expression of Slug is correlated with poor survival. Since Daxx does not regulate the expression of Slug, these data do not show whether Daxx is link to Slug or not. Instead, examining whether Daxx an E-cad expression is inversely correlated would be more revealing.

5. Fig. 7 aims to show hypoxia, a physiological stimulus, could regulate Daxx and invasion. But it is unclear whether endogenous Daxx and Slug play roles in mediating this effect or not. Fig. 7h shows that massive overexpression of Daxx under hypoxia (much higher than Daxx level under normoxia) did very little to E-cad and occludin expression. These data further question the physiological role of Daxx in regulating EMT upon hypoxia.

In summary, although data using overexpression of Daxx somewhat support the model presented in Fig. 8b, all the data with endogenous Daxx shed significant doubt on the physiological significance of the proposed role of Daxx in tumor invasion.

Reviewer #2 (Remarks to the Author)

Lin et al. have nicely shown in the "Daxx inhibits hypoxia-induced lung cancer cell metastasis by suppressing the HIF-1a/HDAC/Slug axis" manuscript a cascade of events where Daxx is the central repressor. As Daxx negatively influences a pathway involved in

cancer invasion, these results have strong interests in the inhibition of cancer metastasis.

The key results are well explained and the manuscript is original and of interest. The statistics are correct.

To improve the strength of this manuscript, I will highly recommend the authors to respond to minor questions:

1- Main concern: the authors have used human siRNA-SMARTpool for Daxx, Slug, HIF-1 α and siControl for all experiments. As in these conditions, less siRNA of each of the 5 siRNA are used, you may expect to decrease the risk of side effects of the siRNA. However, if 1 out of 5 siRNA is linked to a strong side effect, you may under estimate the power of your siRNA and thus the power of interference on your gene. I will thus highly recommend to test 1 or 2 siRNA from the SMARTpool, individually.

2- The authors emphasized in the title about HIF-1 α . However, HIF-1 α appeared for the first time in Figure 7. They seem to mix HIF-1 and HIF-1 α .. What about HIF-2 as HIF-2 is also involved in the EMT?

3- Figure 2d and Figure 8b: the authors are using a "E-cadherin (Ecad) promoter-luciferase reporter construct containing wild-type or mutant forms of the E-box sequence together with Slug and/or Daxx. "

As HIF-1, the transcription factor with both HIF-1 α and HIF-1 β sub-units, recognize also a E-Box sequence, we could expect to obtain a competition in hypoxia in between HIF1 and Slug, even if HIF represses Daxx. The authors should comment on that.

Moreover, what will happen if experiments from Figure 2 were performed in hypoxia?

Figure 8b: the authors meant HIF-1 instead of HIF-1 α ?

4- In the Discussion, the authors said: "We also found that the down-regulation of Daxx under hypoxia was regulated at the transcriptional level by HIF-1 α . However, we did not find a hypoxic response element (HRE) within the Daxx promoter or detect any HIF-1 α -mediated transcriptional-repression ability, suggesting that the down-regulation of Daxx by HIF-1 α occurs through an indirect or post-transcriptional mechanism. Thus, determining how HIF-1 α down-regulates Daxx mRNA is important in uncovering the mechanism of hypoxia-induced cancer metastasis. "

The authors should know that the HRE is not always in the promoter. It has been found in the 3'UTR of EPO and in intron 1 of IGFBP1.

The authors change their conclusions.

5-Occludin : Figure 1c (3 bands), Figure 1f (1 band), Figure 4e (2 bands in CL1-2 cells , and 1 band in CL141 cells), Figure 7g (1 band). The authors should comment.

Moreover, Figure 4e, there is a shift of Occludin when the FL of Daxx is expressed in CL141 cells. The authors should comment.

6- Figure 2c and d: expression of Slug and Daxx should be shown in these figures.

7- Figure 2d: why the E-box mut reporter presented more relative luciferase activity compared to the E-box wt?

8- Figure 4e: actin is overexposed as in many experiments of this manuscript.

9- Figure 6a and b: as the CocL2 showed a strong stabilization of HIF-1 α better than 1% O₂ and thus a better decrease in Daxx expression, did the authors tried experiments at 0.1% O₂?

10- Suppl. Figure S1a and S1b: Daxx expression is totally different in CL1-0. The authors should comment.

11- Suppl. Figure S1b Daxx expression is slightly decreased in CL1 -5 compared to CL1-2 or CL1-0, however, the impact on Slug is strong. The authors should comment.

12- Suppl. S7 and Suppl. S9a seem to be the exact same figure.

Reviewer #3 (Remarks to the Author)

The presented study is well-performed and investigates the molecular mechanisms of hypoxia-induced lung cancer metastasis. The study concludes that Daxx can inhibit hypoxia-induced lung cancer metastasis by attenuating Slug-mediated transcriptional repression of epithelial-like markers that in turn cause cells to exhibit low invasiveness. Mechanistic studies in cell lines investigating the interrelation between Daxx and Slug expression/activity and their role for invasiveness and migration are performed in a selection of lung cancer cell lines. The cellular studies are followed up by in vivo studies using orthotopic lung models and in addition include prognostic/survival analysis of Daxx and Slug expression levels in lung cancer patient tumor specimens. The authors hereafter look into the effects of hypoxia for the expression/functional activity of Daxx an

Slug. These mechanistic studies are in addition followed up by prognostic/ survival analysis of Daxx and HIFalpha expression levels in lung cancer patient tumor material.

A few major points remained to be addressed in the manuscript:

-The authors include a set of lung cancer cell lines that they use for mechanistic studies of the Daxx-Slug axis. As shown in the supplemental data some lung cancer cell lines exhibit epithelial-like phenotype (high E-cadherin, high occludin) that corresponds with expression of Daxx while other cell lines exhibit a mesenchymal-like phenotype that corresponds with expression of Slug (and low/no expression of Daxx). However, the authors do not investigate whether the inverse relationship between Daxx and Slug expression (and epithelial-like versus mesenchymal-like markers) correspond to specific lung cancer genotypes such as mutations of Kras, EGFR and other known oncogenes. Given that the different lung cancer genotypes require different treatment strategies the paper would benefit from including more lung cancer cell lines in supplemental data 1A along with their corresponding genotypes (driver mutations). The implication of driver mutation for the indicated Daxx/Slug phenotypes should be discussed in relation to treatment strategies.

-The mechanistic cell culture studies are confined to a few selected cell lines. Please include 1-2 more cell lines for knockdown and rescue experiments to increase the breadth of the findings in relation to lung cancer.

-For the patient data and prognostic/survival studies it would in addition be beneficial if any genotype status were known. If so, please include and discuss.

-By different ways of hypoxia induction, Daxx protein and mRNA levels decreased and CDH1 and OCLN mRNA levels decreased as well (figure 7a-b, and 7f), and further knockdown HIF1a restored Daxx mRNA level back to normal (figure 7c-d). However, the change of Slug in respond to normoxic or hypoxic was not consistent (figure 7a-b and 7h), minor increased in figure 7a but kept consistent in the figure 7b if we believed your densitometry number. Slug functions as a key mediator/transcription factor in linking Daxx to CDH1/OCLN transcriptional regulation, you need to show how HIF1a, Daxx, and Slug interaction to regulate the transcription of CDH1/OCLN and tumor metastasis.

Point-by-point responses to each of the reviewers' comments of first revision

We are grateful to all of the reviewers for their critical comments and insightful suggestions, which have helped us considerably improve our paper. As indicated in the responses that follow, we have taken all of these comments and suggestions into account in the revised manuscript, including the Supplementary Information.

Reviewers' comments:

Reviewer #1:Expert in metastasis and EMT

The manuscript entitled "Daxx Inhibits Hypoxia-Induced Lung Cancer Cell Metastasis by Suppressing the HIF-1 α /HDAC1/Slug Axis" by Lin et al. identified Daxx as being critical in preventing EMT and invasion in lung tumor cells. Mechanistically, the study suggests that Daxx binds to Slug and inhibits HDAC1 binding to Slug, thus antagonizing the ability of Slug to bind and suppress the E-cadherin promoter. Furthermore, the authors provided some data suggesting that HIF1 α could repress Daxx expression, and overexpression of Daxx can suppress HIF1 α -induced invasion. In vivo, overexpression of Daxx can suppress metastasis. In patients, the HIF1 α , Daxx and Slug pathway is associated with outcome.

In general, identification of Daxx as a suppressor of EMT and tumor invasion is novel and interesting, thus worth further investigation. The association of Daxx with good outcome is striking and interesting. However, the proposed

mechanism linking Daxx and Slug is not well developed. Especially, the physiological significance of endogenous Daxx in suppressing Slug to maintain epithelial identity is not supported by the experimental data.

In summary, although data using overexpression of Daxx somewhat support the model presented in Fig. 8b, all the data with endogenous Daxx shed significant doubt on the physiological significance of the proposed role of Daxx in tumor invasion.

Major points:

[Rev.1- 1] The expression level of Daxx and Slug in epithelial vs. mesenchymal lung cancer cells questions whether endogenous Daxx indeed is required to suppress Slug in epithelial lung cancer cells. Fig.S1a shows that all four epithelial cells express Daxx, but not Slug. Therefore, in these four epithelial tumor cell lines, Daxx is not needed to suppress Slug since Slug is not expressed in these cells. In the three mesenchymal cells presented in Fig. S1a, only CL-141 express both Daxx and Slug, but E-cadherin is completely suppressed, suggesting that Daxx is not able to suppress Slug in CL-141 cells. It is possible that Daxx level is too low in CL-141 cells to suppress Slug. Then comparing the level of Daxx expression in Fig. S1a and Fig. S1b, it is clear that the Daxx level is highest in CL1-0 cells and second highest in CL1-2 and then CL1-5 cells, much higher than H23, H1975 and PC9 cells. But even such high levels of Daxx in CL1-2 and CL1-5 cells could not suppress Slug in these cells to promote E-cadherin expression, thus questioning whether Daxx is truly involved in inhibiting the mesenchymal state. Instead, the E-cadherin status seems to be mostly correlated with Slug expression level in these three cell

lines presented in Fig.1b.

[Answer to Rev.1- 1]

We thank you for your critical comment and apologize that the format of the original Supplementary Figure 1 might have been an issue for the reviewer and readers. We would note that the original expression levels of Daxx, Slug, and mesenchymal/epithelial markers in seven lung cancer cell lines and CL-series cells shown in Supplementary Figure 1a and 1b by Western blotting were relative expression levels, not absolute amounts. Thus, we re-examined Daxx and Slug expression using different exposure times and in additional cell lines, as also suggested by Reviewer #3. The results indicate that, among the 10 cell lines, H1975, PC9, CL100 and H23 cells expressed relatively lower amounts of endogenous Slug, whereas CL1-2, CL141, Hop-92, Hop-62 and H460 cells expressed relatively lower amounts of endogenous Daxx (revised Supplementary Figure 1a). We sincerely apologize for the fact that the list of selected cell lines did not consistently reflect the Daxx-Slug-E-cadherin axis that this manuscript intended to illustrate. Because each cancer cell line has different genetic alterations that may confer diverse oncogenic pathways, the main purpose in performing the cell line screening shown in Supplementary Figure1 was to identify suitable cell lines that might allow us to clarify the relationship between Daxx and Slug, and elucidate their downstream functions.

The cell lines chosen for subsequent experiments were CL1-2, CL1-5, CL141 and PC9, which expressed endogenous Slug, Daxx, E-cadherin, and occludin. These cell lines allowed us to study Daxx and Slug regulatory mechanism by manipulating their expression. Using these cell lines, we showed in subsequent experiments performed in this manuscript that Daxx is involved

in Slug-mediated EMT and invasiveness.

We also re-examined the endogenous expression levels of EMT-related marker proteins in CL1-0, CL1-1, CL1-2, and CL1-5 cells. This analysis showed that Daxx, E-cadherin, and occludin were expressed at relatively low levels in CL1-2 and CL1-5 cells, whereas Slug and N-cadherin were expressed at higher levels in these cells (revised Supplementary Figure 1b). We conclude that Daxx expression correlates with E-cadherin and occludin, and is inversely correlated with cell invasiveness. However, in these experiments, we were not able to quantify the absolute amount of Daxx and Slug protein, which could impact E-cadherin expression.

As suggested by the reviewer, we have modified our description in the Results section of the revised manuscript, as follows:

"We found that endogenous levels of Daxx expression correlated with E-cadherin and occludin and was inversely correlated with cell invasiveness (Supplementary Figure 1a and 1b)." (Manuscript Page 6)

Supplementary Figure 1

[Rev.1- 2(1)] Although data using overexpression of Daxx and Slug support their interaction and the potential role of Daxx in suppressing Slug transcriptional activity, all the data examining endogenous Daxx and Slug shed doubts on the significance of this interaction. For example, Fig. 1C shows that knockdown of Daxx in PC9 cells could downregulate E-cadherin, but this could not be due to Slug since PC9 cells show no Slug expression (Fig.S1a).

[Answer to Rev.1-2(1)]

We again apologize for the possible confusion caused by the original Fig. S1a as well as for our insufficiently clear description in Figure 1C and Supplementary Fig.3b, which was intended to show that PC9 cells did express

Slug, which was not regulated by Daxx (original Supplementary Figure 3b). We have merged Supplementary Figure 3b into Figure 1 (C and F) in the revised manuscript.

Revised Figure 1C

Revised Figure 1F

[Rev.1- 2(2)] For CL1-2 cells, to show that this effect is due to Slug suppression, double knockdown of Slug and Daxx is needed to show that E-cadherin suppression upon Daxx knockdown is due to Slug.

[Answer to Rev.1-2(2)]

We thank you for your insightful comment, which may help reinforce the concept that we attempted to explore in this manuscript. As per your suggestion, we used a double-knockdown approach in both CL1-2 and CL141 cells to study whether endogenous Daxx acts through Slug to play a role in E-cadherin/occludin expression or cell invasiveness. To achieve this goal, we measured mRNA and protein expression levels of E-cadherin and occludin and performed invasion and migration assays. The results showed that double-knockdown of Daxx and Slug (siDaxx+siSlug) abolished the suppressive effects

of siDaxx on CDH1/OCLN mRNA expression. Double-knockdown of Daxx and Slug also abolished siDaxx-induced increases in cell motility and invasiveness. These results support our conclusion that Daxx suppresses cell invasion/motility through Slug-mediated suppression of E-cadherin and occludin expression. We have added these results to our revised manuscript (Figure 4c and 4d and Supplementary Figure 7b).

Revised Figure 4c, 4d and Supplementary Figure 7b

[Rev.1- 2(3)] Fig. 2B intends to show that endogenous Daxx and Slug interact, but the Co-IP signals are very weak with lots of background, therefore not convincing.

[Answer to Rev.1- 2(3)]

Since the interaction between Daxx and Slug in cells could be dynamic, we re-examined the endogenous interaction of Daxx and Slug by Co-IP, including additional treatment with the crosslinking reagent, DSP. The results of these experiments demonstrated association of endogenous Daxx and Slug in CL1-5 cells.

Revised Figure 2b

[Rev.1- 3] Fig. 3, 4, 5 and 7 are largely based on overexpression of Daxx and Slug. Given all the issues discussed above on endogenous Daxx and Slug function, it is unclear whether physiological function of Daxx is indeed what described for overexpressed Daxx. For example, CL1-5 is the most invasive cells discussed in the paper. Instead of overexpressing Slug and Daxx in already invasive CL1-5 cells used in Fig. 5, The more critical experiment is to

test whether removal of endogenous Daxx in CL1-2 cells could promote invasion and metastasis in a Slug-dependent manner.

[Answer to Rev.1- 3]

We thank you for your invaluable suggestion. As you noted, we performed double-knockdown experiments to determine whether Slug is involved in Daxx-mediated suppression of cell invasiveness *in vitro* using an siRNA approach (Figure 4c and 4d). To generate stable knockdown cell lines for application in an *in vivo* animal model, we used lentivirus-based shRNAs against Daxx and Slug (Figure 5e, Supplementary Figure 8a, and 8b). CL1-2 cells stably expressing control shRNA (shCtrl), shDaxx, or shDaxx plus shSlug were orthotopically transplanted into the left lungs of mice. The results showed that CL1-2 cells exhibited weaker tumorigenic and metastatic ability compared with CL1-5 cells, as evidenced by the fact that no primary tumors formed in the control CL1-2 group 40 days after implantation (0%; Figure 5e). However, four mice (44%) transplanted with shDaxx-infected CL1-2 cells generated primary tumors, two of which further formed metastatic nodules. These phenomena were not observed in the shDaxx plus shSlug group (Figure 5e), supporting the interpretation that Daxx acts through Slug to mediate suppression of cell invasiveness.

We also intravenously injected CL1-2 cells in an attempt to detect late-stage metastasis. Due to the available number of NOD-SCID mice in the limited time, we divided the number of mice to 5 in each group. 45 days after tumor cells injection, we harvested mice lung and checked the metastatic nodule(s) (Supplementary Figure 8c). The result showed that shDaxx-expressing CL1-2 cells exhibited malignant metastatic abilities compared to control CL1-2 cells, while additional knockdown of Slug reduced the effects of Daxx-knockdown on

cancer metastasis. These results support the idea that Daxx-mediated suppression of cancer metastasis is dependent upon Slug.

Revised Figure 4d

Revised Figure 5f and supplementary Figure 8

f

	Lung tumor incidence	Metastatic nodule incidence
shCtrl	0% (0/9)	0%(0/9)
shDaxx	44% (4/9)	50% (2/4)
shDaxx+shSlug	0% (0/9)	0% (0/9)

[Rev.1- 4] Fig. 6 shows that expression of Daxx is correlated with good survival, while expression of Slug is correlated with poor survival. Since Daxx does not regulate the expression of Slug, these data do not show whether Daxx is link to Slug or not. Instead, examining whether Daxx an E-cad expression is inversely correlated would be more revealing.

[Answer to Rev.1- 4]

Thank you for your insightful suggestion. We agree that the correlation of Slug expression with poor survival was not caused by Daxx regulation, and E-cadherin expression could reflect Slug activity and is a better indicator. Using an immunohistochemical approach, we measured E-cadherin expression in 83 lung cancer specimens that displayed positive Slug expression. Kaplan-Meier analyses showed that high E-cadherin expression, which might represent low

Slug activity, correlated with good overall survival ($P = 0.037$; Figure 6c). Considering E-cadherin and Daxx expression status together, patients with low Daxx and low E-cadherin expression had the worst overall survival. Conversely, patients with high Daxx and high E-cadherin had better overall survival ($P < 0.001$; Figure 6d and Table 1). These results indicate that the combination of Daxx and E-cadherin expression could be a better prognostic marker for the positive Slug-expressing cohort (Figure 6a). We have added these results to the revised manuscript in Figure 6

Revised Figure 6a, 6c, 6d and Table 1

Revised Table 1.

Table 1

Variable	Hazard Ratio (95% CI)	p-value
Daxx (>10% v.s <10%)		
Daxx	0.202 (0.06- 0.681)	0.0099
Sex	1.59 (0.609-4.15)	0.3435
Cell type	1.282 (0.586-2.805)	0.5339
Stage	1.089 (0.732-1.62)	0.6749
Age	0.989 (0.95-1.03)	0.5896
E-cadherin (>52% v.s. <52%)		
E-cadherin	0.64 (0.32- 1.25)	0.19
Sex	1.86 (0.94-3.68)	0.076
Cell type	0.59 (0.28- 1.25)	0.172
Stage	1.76 (0.79- 3.95)	0.167
Age	0.98 (0.95- 1.25)	0.272
Combine (Daxx⁺/E-cad^{high} v.s Daxx⁻/ E-cad^{low} v.s Others)		
Daxx ⁺ / E-cad ^{high}	0.20 (0.05 - 0.86)	0.031
Others	0.35 (0.17 - 0.72)	0.004
Age	0.99 (0.98 - 1.03)	0.749
Sex	2.00 (0.32 - 3.92)	0.042
Cell type	0.67 (0.32 - 1.41)	0.289
Stage	1.64 (0.75 - 3.58)	0.217

* Stepwise selection was used to select the optimal multivariable Cox proportional hazard regression model. Daxx expression was designated as 'high' or 'low' using 10% cell positivity as the cut-off point, while E-cadherin cut-off point was set as 52% positive expression. The divided cohort was adjusted by age, sex, histological type, and stage. *P*-values (two-sided) were calculated using a chi-square test. CI, confidence interval.

[Rev.1- 5] Fig. 7 aims to show hypoxia, a physiological stimulus, could regulate Daxx and invasion. But it is unclear whether endogenous Daxx and Slug play roles in mediating this effect or not. Fig. 7h shows that massive overexpression of Daxx under hypoxia (much higher than Daxx level under normoxia) did very little to E-cad and occludin expression. These data further question the physiological role of Daxx in regulating EMT upon hypoxia.

[Answer to Rev.1- 5]

We thank you for the critical comment. As you noted, we attempt to study

whether the Daxx/Slug axis mediates hypoxia-induced cell invasion. Initially, we observed that hypoxic stress altered Daxx and Slug expression in lung cancer cells (Figure 7a). Because hypoxia induces numerous signaling pathways in addition to the HIF- α pathway, we treated cells with CoCl₂, a specific HIF- α activator. These experiments showed that CoCl₂ induced up-regulation of HIFs- α accompanied by down-regulation of Daxx and modest up-regulation of Slug (Figure 7b). In addition, knockdown of HIF-1 α , but not HIF-2 α , abolished the hypoxia-induced down-regulation of Daxx mRNA expression without obviously affecting Slug mRNA levels (Figure 7c and 7d, and Supplementary Figure 10a, 10b, and 10c). These results further support the conclusion that hypoxia-induced down-regulation of Daxx expression is mediated by HIF-1 α .

To determine whether hypoxia alters Daxx expression and thereby influences cancer cell invasiveness, we performed *in vitro* invasion assays under hypoxic conditions in cells in which endogenous HIF-1 α , Daxx, and/or Slug were knocked down. These experiments showed that knockdown of HIF-1 α or Slug under hypoxic conditions increased CDH1/OCLN mRNA expression and decreased hypoxia-induced cell invasion (Figure 7e and 7f), indicating that both HIF-1 α and Slug mediate the enhanced cell invasion observed under hypoxic conditions. Moreover, additional knockdown of Daxx partially reversed the effects of HIF-1 α silencing on CDH1 and OCLN expression as well as cell invasion (Figure 7e and 7f). Moreover, re-expression of Daxx attenuated the hypoxia caused down-regulation of E-cadherin and occludin expression (Figure 7g and Supplementary Figure 12a and b), which is also reflected to the attenuated cell invasiveness of Daxx-overexpressing cells under hypoxia (Figure 7h and Supplementary Figure 12c). In addition, knockdown of Slug

under hypoxia accompany with Daxx overexpression displayed similar effects to Daxx over-expression alone (Figure 7g and 7h). Collectively, these results suggest that HIF-1 α regulates cancer cell invasion upon hypoxia exposure by suppressing Daxx expression through its inability of blocking Slug activation.

We found that Daxx regulates EMT mainly through a Slug-dependent pathway, so that the impact of Daxx overexpression on the up-regulation of CDH1 and OCLN is dependent upon Slug expression levels. Although we found the expression levels of Slug were elevated under hypoxia, the hypoxic effect on Slug expression was not substantial. This probably explains why massive overexpression of Daxx under hypoxia did very little to E-cad and occludin expression. To avoid confusing the readers, we revisiting this experiment using a lower level of Daxx overexpression, we found a similar effect on E-cadherin and occludin recovery (Supplementary Figure 12b).

Revised Figure 7

Revised Supplementary Figure 10

Revised Supplementary Figure 12

S12a

S12b

S12c

Reviewer #2:Expert in hypoxia

Lin et al. have nicely shown in the "Daxx inhibits hypoxia-induced lung cancer cell metastasis by suppressing the HIF-1a/HDAC/Slug axis" manuscript a cascade of events where Daxx is the central repressor. As Daxx negatively influences a pathway involved in cancer invasion, these results have strong interests in the inhibition of cancer metastasis.

The key results are well explained and the manuscript is original and of interest.

The statistics are correct.

To improve the strength of this manuscript, I will highly recommend the authors to respond to minor questions:

[Rev.2- 1] Main concern: the authors have used human siRNA-SMARTpool for Daxx, Slug, HIF-1 α and siControl for all experiments. As in these conditions, less siRNA of each of the 5 siRNA are used, you may expect to decrease the risk of side effects of the siRNA. However, if 1 out of 5 siRNA is linked to a strong side effect, you may under estimate the power of your siRNA and thus the power of interference on your gene.

I will thus highly recommend to test 1 or 2 siRNA from the SMARTpool, individually.

[Answer to Rev.2- 1]

We thank you for your critical assessment. Since off-target effects are always a concern with siRNA approaches, we used a SMARTpool for gene silencing, based on their lowest off-target effect compared with other commercial products (1). However, in response to your valuable suggestion, we tested two individual siRNAs in the siRNA pool. The results obtained confirmed that knockdown of Daxx using either pooled siRNAs or single siRNAs (siDaxx-1 and siDaxx -2)

effectively down-regulated E-cadherin and occludin protein levels and invasion-suppressing activity (Supplementary Figure 2a and 2b). The effect of single strand siSlug siRNAs (siSlug-1 and siSlug-2) also acts like pooled siRNA (siSlug-pool), which elevated E-cadherin and occludin expression (Supplementary Figure 2c). Moreover, by examining endogenous Daxx mRNA expression using individual HIF1 α siRNAs, we found that each HIF-1 α siRNA up-regulated Daxx expression (Supplementary Figure 10a).

Revised Supplementary Figure 2 and 10a

[Rev.2- 2] The authors emphasized in the title about HIF-1 α . However, HIF-1 α appeared for the first time in Figure 7. They seem to mix HIF-1 and HIF-1 α .

What about HIF-2 as HIF-2 is also involved in the EMT?

[Answer to Rev.2- 2]

We appreciate the reviewer's pivotal suggestion. First, we apologize for not clearly describing HIF-1 and HIF1- α . We have corrected this in the revised manuscript, noting that the HIF-1 complex consists of HIF-1 α and HIF-1 β , but only HIF-1 α is stabilized under hypoxic conditions. In the revised manuscript (Figure 7a and 7b), we examined the expression of both HIF-1 β and HIF-2 α , showing that HIF-1 β was constantly expressed in CL1-2 and PC9 cells under both normoxic and hypoxic condition, whereas HIF-2 α expression was elevated under hypoxic conditions (Figure 7a and 7b). Although both HIF-1 α and HIF-2 α levels were inversely correlated with Daxx under hypoxic conditions or following treatment with CoCl₂ (Figure 7a. and 7b), Daxx mRNA expression was only up-regulated following silencing of HIF-1 α (Figure 7c.). This strongly suggests that HIF-1 α but not HIF-2 α suppresses Daxx expression under hypoxic conditions. We also added an additional sentence in the first paragraph of the revised Introduction to introduce the function of the HIF1/2 complex under hypoxic conditions.

Revised Figure 7a-c

Revised Manuscript Page 4:

Under hypoxic conditions, the hypoxia-inducible factors (HIFs), HIF-1 α and HIF-2 α , are stabilized, enabling them to coordinately regulate the expression of genes required for promoting disseminated, invasive and angiogenic properties, shifting cancer cells towards a metastatic phenotype (3,4). Specifically, hypoxia-stabilized HIF-1 α has been shown to up-regulate epithelial-mesenchymal transition (EMT)-related transcription factors (EMT-TFs), including TWIST and Snail, indicating that HIF-1 α plays a critical role in hypoxia-induced EMT (5, 6).

[Rev.2- 3(1)] Figure 2d and Figure 8b: the authors are using a "E-cadherin (Ecad) promoter-luciferase reporter construct containing wild-type or mutant forms of the E-box sequence together with Slug and/or Daxx." As HIF-1, the transcription factor with both HIF-1 α and HIF-1 β sub-units, recognize also a E-Box sequence, we could expect to obtain a competition in hypoxia in between HIF1 and Slug, even if HIF represses Daxx. The authors should comment on that.

[Answer to Rev.2- 3(1)]

We thank you for your constructive comment. To the best of our knowledge, the E-box consensus is minimally defined as CANNTG, yet the adjacent nucleotides of E-boxes are variable for genes regulated by bHLH and Snail/Slug-related proteins (2, 3). According to a study published by *Hagra., et al.*, Slug could recognize three E-boxes (EboxA-C) located in the E-cadherin promoter. Among these E-boxes, EboxC (CACCTG) appeared to play the most significant role in the repression of E-cadherin gene transcription (4). Another study indicated the E-cadherin promoter contains eight hypoxia response elements (HRE; -CGTG-) and one ARNT/HIF-1 β binding site, which is distinct from known E-box locations (5). It also indicated that HIF-1 α transactivates E-cadherin expression in an E-box-independent manner, showing that a luciferase reporter construct containing only minimal E-box regions is ineffective in mediating hypoxia-dependent expression(5). However, we cannot exclude the possibility that HIF-1 binds directly to the E-box. By overexpressing a stable HIF-1 α construct that could not be degraded under normoxia (HIF-1 α - Δ ODD), we found that HIF-1 α did not affect an E-cadherin promoter that contains wild-type E-boxes (Figure Rev. 2a). Taken together, these findings

suggest that HIF-1 does not compete with Slug for E-boxes within the E-cadherin promoter under hypoxic conditions. However, it is possible that HIF-1 could affect E-cadherin expression through interactions with other target sites.

Figure Rev. 2a

[Rev.2- 3(2)] Moreover, what will happen if experiments from Figure 2 were performed in hypoxia?

[Answer to Rev.2- 3(2)]

We performed E-cadherin reporter assays as originally shown in Figure 2d under hypoxic conditions in CL1-2 cells (Figure Rev.2b). These experiments showed that hypoxia had very little effect on E-cadherin reporter activity.

Figure Rev.2b

[Rev.2- 3(3)] Figure 8b: the authors meant HIF-1 instead of HIF-1 α ?

[Answer to Query No.3-3]

Thank you for your attention to detail. We have changed HIF-1 α to HIF-1 in the revised Figure 8b.

[Rev.2- 4] In the Discussion, the authors said: "We also found that the down-regulation of Daxx under hypoxia was regulated at the transcriptional level by HIF-1 α . However, we did not find a hypoxic response element (HRE) within the Daxx promoter or detect any HIF-1 α -mediated transcriptional-repression ability, suggesting that the down-regulation of Daxx by HIF-1 α occurs through an indirect or post-transcriptional mechanism. Thus, determining how HIF-1 α down-regulates Daxx mRNA is important in uncovering the mechanism of hypoxia-induced cancer metastasis. "

The authors should know that the HRE is not always in the promoter. It has been found in the 3'UTR of EPO and in intron 1 of IGFBP1.

The authors change their conclusions.

[Answer to Rev.2- 4]

We thank you for your invaluable suggestions, which have improved the quality of our manuscript. We have revised our discussion as follows: " Although we did not find a hypoxic response element (HRE) within the Daxx promoter, we cannot exclude the possibility that HIF-1 α suppresses Daxx mRNA at transcriptional or post-transcriptional levels, because HIF-1 α has been found to regulate EPO[Erythropoietin] through its 3'-UTR and IGFBP1 (insulin-like growth factor-binding protein 1) through its first intron (56) Thus, determining how HIF-1 α down-regulates Daxx expression is important in uncovering the

mechanism of hypoxia-induced cancer metastasis.” (Revised manuscript Page 19)

[Rev.2- 5] Occludin : Figure 1c (3 bands), Figure 1f (1 band), Figure 4e (2 bands in CL1-2 cells , and 1 band in CL141 cells), Figure 7g (1 band). The authors should comment. Moreover, Figure 4e, there is a shift of Occludin when the FL of Daxx is expressed in CL141 cells. The authors should comment.

[Answer to Rev.2- 5]

We thank you for bringing this issue to our attention and apologize for our inconsistency. Occludin consists of α - and β -occludin isoforms, the predicted molecular weight of occludin is around 65 kDa as DeMaio et al described in 2001(Ref. 6, Figure 2a). However, using an siRNA-mediated occludin knockdown approach, we found that all bands identified in the control group were diminished (Figure Rev.2c). The available information suggests the presence of different isoforms reflecting posttranscriptional and/or posttranslational modifications(7); whereas the changing proportions may have resulted from differences in the stringency of immunoblot lysis buffers (without phosphatase inhibitors). Although all visible bands were diminished upon treatment with occludin siRNA, in our revised manuscript, we focused on two bands that displayed the strongest signals and were close to the calculated protein molecular weight (*) so as not to mislead readers. Notably, these two bands sometimes appeared as a single band with longer exposure times, possibly explaining why some of our blots showed only one band.

We cannot exclude the possibility that overexpression of full-length Daxx in CL-141 cells might induce the expression of a specific occludin isoform or posttranslational modifications of occludin.

Figure Rev.2c

[Rev.2- 6] Figure 2c and d: expression of Slug and Daxx should be shown in these figures.

[Answer to Rev.2- 6]

In response to the reviewer's helpful suggestion, we have added Western blotting results to the revised Figure 2c and 2d.

Revised Figure 2c and 2d

[Rev.2- 7] Figure 2d: why the E-box mut reporter presented more relative luciferase activity compared to the E-box wt?

[Answer to Rev.2- 7]

We apologize for not providing a sufficient explanation in Figure 2d. Our conclusion is that there are several transcriptional repressors, including Snail/Slug, Twist, ZEB1/2 and E47, which transcriptionally repress E-cadherin expression through E-box binding. The construct containing mutant E-boxes therefore would have higher basic luciferase activity if the cells we used contained any of the above-mentioned transcriptional repressors. In order to avoid misunderstanding, we normalized results for the wild-type construct and mutant construct to their respective controls in the revised Figure 2d.

Revised Figure 2d

[Rev.2- 8] Figure 4e: actin is overexposed as in many experiments of this manuscript.

[Answer to Rev.2- 8]

Thank you for the critical assessment. In response, we re-exposed blots for actin expression in the revised figures

[Rev.2- 9] Figure 6a and b: as the CoCl₂ showed a strong stabilization of HIF-1 α better than 1% O₂ and thus a better decrease in Daxx expression, did the authors tried experiments at 0.1% O₂?

[Answer to Rev.2- 9]

We appreciate your constructive suggestion. We did test other O₂ levels over the hypoxia range (0-10% (8)), but ended up choosing 2% O₂ as the hypoxic condition in this study because we found that the lung cancer cell line we used was unhealthy when incubated in 1% O₂ (data not shown). Although the hypoxia-stabilized activity of HIF-1 α under 2% O₂ was not stronger than that observed with CoCl₂ treatment, it more likely mimics the *in vivo* microenvironment.

[Rev.2- 10] Suppl. Figure S1a and S1b: Daxx expression is totally different in CL1-0. The authors should comment.

[Answer to Rev.2- 10]

We apologize for not providing sufficient information in Figure S1a and S1b, a point also raised by Reviewer #1. Compared with other cell lines, the CL1 series has relatively higher Daxx expression; hence, we showed a Daxx blot

with a longer exposure time in original Figure S1a. To improve our assessment of Daxx protein expression, we present better quality blots in revised Figure S1a and S1b and show protein expression at two exposure times. We have included results from additional cell lines in the revised Figure S1a and S1b, as also suggested by Reviewer #3 (see below).

Supplementary Figure 1a and 1b

[Rev.2- 11] Suppl. Figure S1b Daxx expression is slightly decreased in CL1 - 5 compared to CL1-2 or CL1-0, however, the impact on Slug is strong. The authors should comment.

[Answer to Rev.2- 11]

Thank you for your critical assessment. We reported on the Slug–E-cadherin regulatory axis in our previous studies, showing that Slug has a strong impact

on E-cadherin expression in CL1-series cell lines (9, 10). Therefore, although Daxx and E-cadherin expression were gradually down-regulated in CL1-2 and CL1-5 lines compared to CL1-0, Slug may still play a major role in controlling E-cadherin expression. Since Daxx plays as a upstream regulator of Slug, the expression of E-cadherin as well as cell invasiveness could be dependent on both Daxx and Slug expressions (Figure S1b).

[Rev.2- 12] Suppl. S7 and Suppl. S9a seem to be the exact same figure.

[Answer to Rev.2- 12]

We thank you for bringing this issue to our attention and apologize for not clearly distinguishing the differences between these figures. As you correctly surmised, Figures S7 and S9 came from the same ChIP assay; thus, the Input and IgG control panels were identical. However, in Figure S7, we demonstrated the binding of Slug and Daxx to the CAR promoter using their corresponding antibodies, whereas in Figure S9, we demonstrated the binding activity of HDAC1 on the CAR promoter. We apologize again for dividing one experiment into two parts. We have combined the original Figures S7 and S9 into a single figure (Supplementary Figure 4) in the revised manuscript.

Supplementary Figure 4

Reviewer #3 (Remarks to the Author): Expert in in lung cancer

The presented study is well-performed and investigates the molecular mechanisms of hypoxia-induced lung cancer metastasis. The study concludes that Daxx can inhibit hypoxia-induced lung cancer metastasis by attenuating Slug-mediated transcriptional repression of epithelial-like markers that in turn cause cells to exhibit low invasiveness. Mechanistic studies in cell lines investigating the interrelation between Daxx and Slug expression/activity and their role for invasiveness and migration are performed in a selection of lung cancer cell lines. The cellular studies are followed up by in vivo studies using orthotopic lung models and in addition include prognostic/survival analysis of Daxx and Slug expression levels in lung cancer patient tumor specimens. The authors hereafter look into the effects of hypoxia for the expression/functional activity of Daxx an Slug. These mechanistic studies are in addition followed up

by prognostic/ survival analysis of Daxx and HIFalpha expression levels in lung cancer patient tumor material.

A few major points remained to be addressed in the manuscript:

[Rev.3- 1]The authors include a set of lung cancer cell lines that they use for mechanistic studies of the Daxx-Slug axis. As shown in the supplemental data some lung cancer cell lines exhibit epithelial-like phenotype (high E-cadherin, high occludin) that corresponds with expression of Daxx while other cell lines exhibit a mesenchymal-like phenotype that corresponds with expression of Slug (and low/no expression of Daxx). However, the authors do not investigate whether the inverse relationship between Daxx and Slug expression (and epithelial-like versus mesenchymal-like markers) correspond to specific lung cancer genotypes such as mutations of Kras, EGFR and other known oncogenes. Given that the different lung cancer genotypes require different treatment strategies the paper would benefit from including more lung cancer cell lines in supplemental data 1A along with their corresponding genotypes (driver mutations). The implication of driver mutation for the indicated Daxx/Slug phenotypes should be discussed in relation to treatment strategies.

[Answer to Rev.3- 1]

Thank you for your insightful suggestion. We have included additional lung cancer cell lines and added their EGFR and Kras mutation status to the revised Supplementary Figure 1a. However, we found that these mutations do not correlate with a Daxx/Slug axis-related phenotype (i.e., cell invasiveness or EMT), suggesting that the Daxx/Slug axis may act through an oncogenic pathway distinct from EGFR and Kras driver mutations.

Revised Supplementary Figure 1a

[Rev.3- 2] The mechanistic cell culture studies are confined to a few selected cell lines. Please include 1-2 more cell lines for knockdown and rescue experiments to increase the breadth of the findings in relation to lung cancer.

[Answer to Rev.3- 2]

We thank you for your helpful suggestion. In response, we have added results from Daxx-knockdown functional assays in the additional cell line, PC9, in Figure 1b and Figure 4b. We also performed Daxx and Slug double-knockdown assays to assess whether endogenous Daxx regulates E-cadherin and occludin expression as well as cell invasiveness through Slug in CL1-2 and

CL-141 cells (Figure 4c and 4d). In the Daxx and Slug rescue experiments, we also confirmed our results in CL141 and HEK-293 cells in addition to CL1-2 cells (Figure 4f and Supplementary Figure 7c). The role of Daxx in regulating hypoxia-mediated EMT and cell invasion was additionally studied in CL1-2 and PC9 cells (Figure 7 and Supplementary Figure 12).

[Rev.3- 3] For the patient data and prognostic/survival studies it would in addition be beneficial if any genotype status were known. If so, please include and discuss.

[Answer to Rev.3- 3]

We thank you for your invaluable suggestion. We validated the genetic status of EGFR and Kras in 76 out of 83 clinical cohort samples studied in this manuscript (the remaining FFPE sample was lost in seven cases) using MALDI-TOF MS assay (11). Cancer-driver mutations in EGFR, KRAS, HER2 and BRAF genes were identified in 37 (48.6%), 5 (6%), 3 (3.9%) and 0 (0%) patients, respectively. Since EGFR has a much higher mutation rate in the general population, we tested the relationship of EGFR mutation status with Daxx, Slug, and/or E-cadherin expression using a Fisher's Exact Test (Supplementary Table 4). However, we found no significant correlation between EGFR mutation status and expression of Daxx and/or E-cadherin, as well as Slug. Our data suggest that the Daxx/Slug axis may play a role distinct from that of EGFR, KRAS, HER2, and BRAF mutations in lung cancer progression. We have added this results in supplementary Table 4 and discussed in the Discussion section.

Supplementary Table 4

		EGFR mutation status		
		Positive (n=37)	Negative(n=39)	P value*
Daxx	low	29 (50%)	29 (50%)	0.7896
	high	8 (44.44%)	10 (55.56%)	
Slug	low	9 (34.62%)	17 (65.38%)	0.094
	high	28 (56%)	22 (44%)	
E-cadherin	low	15 (39.47%)	23 (60.53%)	0.1681
	high	22 (57.89%)	16 (42.11%)	
Daxx/ E-cadherin	Daxx-/ E-cad-	14 (45.16%)	17 (54.84%)	0.6073
	Daxx+/ E-cad +	7 (63.64%)	4 (36.36%)	
	Others	16 (47.06%)	18 (52.94%)	

*p-values were calculated by Fisher's Exact Test

[Rev.3- 4]By different ways of hypoxia induction, Daxx protein and mRNA levels decreased and CDH1 and OCLN mRNA levels decreased as well (figure 7a-b, and 7f), and further knockdown HIF1a restored Daxx mRNA level back to normal (figure 7c-d). However, the change of Slug in respond to normoxic or hypoxic was not consistent (figure 7a-b and 7h), minor increased in figure 7a but kept consistent in the figure 7b if we believed your densitometry number. Slug functions as a key mediator/transcription factor in linking Daxx to CDH1/OCLN transcriptional regulation, you need to show how HIF1a, Daxx, and Slug interaction to regulate the transcription of CDH1/OCLN and tumor metastasis.

[Answer to Rev.3- 4]

Thank you for your insightful assessment. We have re-examined the change of Slug in response to normoxic/ hypoxic conditions and CoCl₂ treatment, please check the revised figure 7a and 7b. We found that Slug protein level could be up-regulated under hypoxia, as CoCl₂ treatment only modestly increased Slug protein expression. However, Slug mRNA didn't affected by hypoxic condition as well as in HIF-1/2 α knockdown conditions (Supplementary Figure 10b and 10c), suggesting that hypoxia induced Slug expression is independent of HIFs.

We apologize for not sufficiently explaining and assessing the function of Slug in hypoxia-mediated metastasis. To determine whether Slug is involved in hypoxia-induced cell invasion, we knocked down Slug under hypoxic conditions. We found that the increased invasiveness and CDH1/OCLN down-regulation caused by hypoxia was abolished in the Slug-silenced group (Figure 7e and 7f). In addition, cells re-expression of Daxx accompanied with Slug silencing under hypoxia displayed similar effects to Slug-silenced alone (Figure 7g and 7h), suggesting Daxx functions as an EMT and invasion suppressor mainly through regulating Slug.

On the other hand, knockdown of HIF-1 α abolished hypoxia-induced down-regulation of CDH1/OCLN and concomitant down-regulation of cancer cell invasion, while additional knockdown of Daxx eliminated the HIF-1 α silencing effects (Figure 7e and 7f). Furthermore, re-expression of Daxx attenuated the hypoxia caused down-regulation of E-cadherin and occludin expression (Figure 7g and Supplementary Figure 12a and b), which is also reflected to the attenuated cell invasiveness of Daxx-overexpressing cells under hypoxia (Figure 7h and Supplementary Figure 12c). Collectively, these results suggest that HIF-1 α -mediated Daxx down-regulation contributes to hypoxia-induced

cell dissemination, invasion, and metastasis of lung cancer cells through its inability of blocking Slug activation.

Revised Supplementary Figure 10

Revised Figure 7

Revised Supplementary Figure 11b

S.11b

Revised Supplementary Figure 12

S12a

S12b

S12c

References:

1. Baum P, Fundel-Clemens K, Kreuz S, Kontermann RE, Weith A, Mennerich D, and Rippmann JF. Off-target analysis of control siRNA molecules reveals important differences in the cytokine profile and inflammation response of human fibroblasts. *Oligonucleotides*. 2010;20(1):17-26.
2. Cano A, Perez-Moreno MA, Rodrigo I, Locascio A, Blanco MJ, del Barrio MG, Portillo F, and Nieto MA. The transcription factor snail controls epithelial-mesenchymal transitions by repressing E-cadherin expression. *Nat Cell Biol*. 2000;2(2):76-83.
3. Ikenouchi J, Matsuda M, Furuse M, and Tsukita S. Regulation of tight junctions during the epithelium-mesenchyme transition: direct repression of the gene expression of claudins/occludin by Snail. *J Cell Sci*. 2003;116(Pt 10):1959-67.
4. Hajra KM, Chen DY, and Fearon ER. The SLUG zinc-finger protein represses E-

- cadherin in breast cancer. *Cancer Res.* 2002;62(6):1613-8.
5. Maroni P, Matteucci E, Drago L, Banfi G, Bendinelli P, and Desiderio MA. Hypoxia induced E-cadherin involving regulators of Hippo pathway due to HIF-1alpha stabilization/nuclear translocation in bone metastasis from breast carcinoma. *Exp Cell Res.* 2015;330(2):287-99.
 6. DeMaio L, Chang YS, Gardner TW, Tarbell JM, and Antonetti DA. Shear stress regulates occludin content and phosphorylation. *Am J Physiol Heart Circ Physiol.* 2001;281(1):H105-13.
 7. Cummins PM. Occludin: one protein, many forms. *Mol Cell Biol.* 2012;32(2):242-50.
 8. Araneda OF, and Tuesta M. Lung oxidative damage by hypoxia. *Oxid Med Cell Longev.* 2012;2012(856918).
 9. Wang SP, Wang WL, Chang YL, Wu CT, Chao YC, Kao SH, Yuan A, Lin CW, Yang SC, Chan WK, et al. p53 controls cancer cell invasion by inducing the MDM2-mediated degradation of Slug. *Nat Cell Biol.* 2009;11(6):694-704.
 10. Shih JY, and Yang PC. The EMT regulator slug and lung carcinogenesis. *Carcinogenesis.* 2011;32(9):1299-304.
 11. Su KY, Chen HY, Li KC, Kuo ML, Yang JC, Chan WK, Ho BC, Chang GC, Shih JY, Yu SL, et al. Pretreatment epidermal growth factor receptor (EGFR) T790M mutation predicts shorter EGFR tyrosine kinase inhibitor response duration in patients with non-small-cell lung cancer. *J Clin Oncol.* 2012;30(4):433-40.

Reviewer #1 (Remarks to the Author):

The revised manuscript has addressed the main concern on whether endogenous Daxx regulates Slug-mediated transcription. Although in most cases, the impact on E-cadherin upon knocking down Daxx is less than 2-fold, the newly added data consistently show that such impact could be biologically significant in regulating invasion and migration.

The following two points should be further addressed.

1. Previous point #4 asked whether Daxx and E-cadherin expression is inversely correlated in patient samples, especially in Slug+ samples. Although the revision stated that Daxx and E-cadherin levels are examined, the correlation data are not presented to address this point, which could significantly strengthen the biological significance of the proposed model.
2. Throughout the study, the hypothesis is that Daxx suppresses Slug-mediated E-cadherin suppression, therefore suppressing EMT and invasion. However, the newly added in vivo experiment in this revision shows that Daxx knockdown cells failed to form primary tumor, suggesting that Daxx likely has some function in regulating primary tumor formation. Whether this tumor formation defect is due to its regulation of Daxx is unclear and needs to be addressed.

Reviewer #2 (Remarks to the Author):

The authors clearly answered all the questions asked by the three referees. The article is now clearer, more consistent, better built and strengthened by the added experiments.

I now accept this revised paper for publication in Nature Communications

Reviewer #3 (Remarks to the Author):

A. The presented study is well-performed and investigates the molecular mechanisms of hypoxia-induced lung cancer metastasis. The study concludes that Daxx can inhibit hypoxia-induced lung cancer metastasis by attenuating Slug-mediated transcriptional repression of epithelial-like markers that in turn cause cells to exhibit low invasiveness. Mechanistic studies in cell lines investigating the interrelation between Daxx and Slug expression/activity and their role for invasiveness and migration are performed in a selection of lung cancer cell lines. The cellular studies are followed up by in vivo studies using orthotopic lung models and in addition include prognostic/survival analysis of Daxx and Slug expression levels in lung cancer patient tumor specimens. The authors hereafter look into the effects of hypoxia for the expression/functional activity of Daxx and Slug. These mechanistic studies are in addition followed up by prognostic/ survival analysis of Daxx and HIF α expression levels in lung cancer patient tumor material. The robustness of findings and conclusions have significantly improved upon revision.

B. The research is original and of interest to the lung cancer field

C. Data / Methodology OK

D. OK

E. Conclusions and robustness have improved after revision. Recommend for publication

F. No further revisions needed.

G. OK

H. OK

Point-by-point responses to each of the reviewers' comments of second revision

Reviewers' comments:

Reviewer #1 (Remarks to the Author):

The revised manuscript has addressed the main concern on whether endogenous Daxx regulates Slug-mediated transcription. Although in most cases, the impact on E-cadherin upon knocking down Daxx is less than 2-fold, the newly added data consistently show that such impact could be biologically significant in regulating invasion and migration.

The following two points should be further addressed.

1. Previous point #4 asked whether Daxx and E-cadherin expression is inversely correlated in patient samples, especially in Slug⁺ samples. Although the revision stated that Daxx and E-cadherin levels are examined, the correlation data are not presented to address this point, which could significantly strengthen the biological significance of the proposed model.

[Answer to Q1] Thank you for your constructive suggestion. We add a tree diagram as we have done in our previous publication¹ to illustrate the preferential expression profile and conditional probabilities of Daxx, Slug, and E-cad expressions in supplementary Figure 9b. The preferential expression patterns of Daxx⁺-Slug^{high} or Daxx⁻-Slug^{high} was found in 60% of patients. In addition, 67% of patients with E-cad^{high} found in Daxx⁺ -Slug^{high} group, while 57% of Daxx⁻ -Slug^{high} showed E-cad^{low}. It demonstrated that Daxx expression level may impact on the expression preference of E-cadherin in Slug^{high} group (Supplementary Figure 9).

The correlation analysis did not show significant direct correlation between Daxx and E-cadherin by IHC. However, we did find a trend of increased positive correlation between Daxx and E-cadherin expression levels in Daxx -high expression samples by Spearman's Rank Correlation (Figure Rev1).

There may be several reasons to explain this result. In this study we showed that Daxx acts as a repressor through HIF-1 α /HDAC1/Slug axis to control Slug- and E-cadherin-mediated cancer cell invasion and correlates with clinical outcome. We did not imply that Daxx directly could regulate E-cadherin

or the Slug-E-cadherin pathway is exclusively controlled by Daxx, as our lab and others showed previously that there are other mechanisms that could regulate Slug and E-cadherin expressions at transcriptional and post-transcriptional level in NSCLC¹⁻⁵. For the reason of cancer complexity and limited case number of Daxx positive population, we may not be able to expect a significant direct positive correlation between Daxx and E-cadherin in this cohort.

Nevertheless, using E-cadherin expression to partially represent Slug transcriptional activity in the Slug-positive NSCLC cohort (we thank you for the constructive suggestion), we first showed that either Daxx-positive expression (Daxx⁺; n=20) or E-cadherin-high expression (E-cad^{high}; n=40) correlates with better overall survival of NSCLC patients (Figure 6c and 6e). Most importantly, combination of Daxx and E-cadherin expressions could further stratify patients' survival in Slug-positive patients (Figure 6f), supporting the idea that Daxx may play a role in lung cancer progression regulated by Slug. It may provide a supporting data for considering that Daxx expression could serve as a prognostic indicator in Slug-positive patients.

Supplementary Figure 9

Supple. Figure 9. The preferential expression patterns of Daxx, Slug, and E-cad represent in a tree diagram. Numbers along edges indicated conditional probabilities of choosing nodes in a path. Color red codes high expression and green codes low expression.

Figure Rev1. Correlations between expressions of Daxx and E-cadherin. X axis represented expression levels of Daxx (staining percentage of cells) and Y axis represented correlation coefficient generated from Spearman's rank correlation. P values of correlation coefficients from large to small were represented from red to blue.

Figure 6b-6f

Q2. Throughout the study, the hypothesis is that Daxx suppresses Slug-mediated E-cadherin suppression, therefore suppressing EMT and invasion. However, the newly added *in vivo* experiment in this revision shows that Daxx knockdown cells failed to form primary tumor, suggesting that Daxx likely has some function in regulating primary tumor formation. Whether this tumor formation defect is due to its regulation of Daxx is unclear and needs to be addressed.

[Answer to Q2] We thank your comprehensive comment. As we performed Daxx and/ or Slug knockdown experiments in both *in vivo* orthotopic lung tumor growth and tail-vein injection assays, Daxx knockdown (shDaxx) group did

display increased tumor formation and metastatic properties (Figure 5f and Supplementary Figure 8c). Moreover, additional knockdown of Slug rescued Daxx silencing effects, suggesting that Daxx could regulate tumor formation and metastasis through modulating Slug. It has been shown that Slug not only regulates EMT but also tumor- initiating ability in several type of cancer⁶⁻⁸. Hence it is possible that Slug may also involve in primary lung tumor formation, which may explain that double knockdown of Daxx and Slug abolished Daxx silencing- caused primary tumor formation and cancer metastasis.

Figure 5f and Supplementary Figure 8c

5f

	Lung tumor incidence	Metastatic nodule incidence
shCtrl	0% (0/9)	0%(0/9)
shDaxx	44% (4/9)	50% (2/4)
shDaxx+shSlug	0% (0/9)	0% (0/9)

S8c

References:

1. Wang, S.P. et al. p53 controls cancer cell invasion by inducing the MDM2-mediated degradation of Slug. *Nat Cell Biol* 11, 694-704 (2009).
2. Shih, J.Y. & Yang, P.C. The EMT regulator slug and lung carcinogenesis. *Carcinogenesis* 32, 1299-1304 (2011).
3. Wu, Y.Y. et al. SCUBE3 is an endogenous TGF-beta receptor ligand and regulates the epithelial-mesenchymal transition in lung cancer. *Oncogene* 30, 3682-3693 (2011).
4. Wu, D.W. et al. FHIT loss confers cisplatin resistance in lung cancer via the AKT/NF-kappaB/Slug-mediated PUMA reduction. *Oncogene* 34, 2505-2515 (2015).
5. Kao, S.H. et al. GSK3beta controls epithelial-mesenchymal transition and tumor metastasis by CHIP-mediated degradation of Slug. *Oncogene* 33, 3172-3182 (2014).
6. Guo, W. et al. Slug and Sox9 cooperatively determine the mammary stem cell state. *Cell* 148, 1015-1028 (2012).
7. Samanta, S. et al. IMP3 promotes stem-like properties in triple-negative breast cancer by regulating SLUG. *Oncogene* 35, 1111-1121 (2016).
8. Luanpitpong, S. et al. SLUG is required for SOX9 stabilization and functions to promote cancer stem cells and metastasis in human lung carcinoma. *Oncogene* 35, 2824-2833 (2016).

Reviewer #2 (Remarks to the Author):

The authors clearly answered all the questions asked by the three referees. The article is now clearer, more consistent, better built and strengthened by the added experiments.

I now accept this revised paper for publication in Nature Communications

Reviewer #3 (Remarks to the Author):

A. The presented study is well-performed and investigates the molecular mechanisms of hypoxia-induced lung cancer metastasis. The study concludes that Daxx can inhibit hypoxia-induced lung cancer metastasis by attenuating Slug-mediated transcriptional repression of epithelial-like markers that in turn cause cells to exhibit low invasiveness. Mechanistic studies in cell lines investigating the interrelation between Daxx and Slug expression/activity and their role for invasiveness and migration are performed in a selection of lung

cancer cell lines. The cellular studies are followed up by in vivo studies using orthotopic lung models and in addition include prognostic/survival analysis of Daxx and Slug expression levels in lung cancer patient tumor specimens. The authors hereafter look into the effects of hypoxia for the expression/functional activity of Daxx and Slug. These mechanistic studies are in addition followed up by prognostic/ survival analysis of Daxx and HIFalpha expression levels in lung cancer patient tumor material. The robustness of findings and conclusions have significantly improved upon revision.

B. The research is original and of interest to the lung cancer field

C. Data / Methodology OK

D. OK

E. Conclusions and robustness have improved after revision. Recommend for publication

F. No further revisions needed.

G. OK

H. OK